# LEARNING DICTIONARIES OVER DATASETS THROUGH WASSERSTEIN BARYCENTERS

## ABSTRACT

Unsupervised Domain Adaptation is an important machine learning problem that aims at mitigating data distribution shifts, when transferring knowledge from one labeled domain to another similar and unlabeled domain. Optimal transport has been shown in previous works to be a powerful tool for comparing and matching empirical distributions. As such, we propose a novel approach for Multi-Source Domain adaptation which consists on learning a Wasserstein Dictionary of labeled empirical distributions, as a means of interpolating distributional shift across several related domains, and inferring labels on the target domain. We evaluate this method on Caltech-Office 10 and Office 31 benchmarks, where we show that our method improves the state-of-the-art of 1.96% and 2.70% respectively. We provide further insight on our dictionary, exploring how interpolations of atoms provide useful predictors for target domain data, and how it can be used to study the geometry of data distributions. Our framework opens interesting perspectives for fitting and generating datasets based on learned probability distributions.

## 1 INTRODUCTION

Dictionary Learning (DiL) is a representation learning technique that seeks to express a set of vectors in $\mathbb{R}^d$ as the linear weighted combinations of elementary elements named atoms. When vectors represent histograms, this problem is known as Nonnegative Matrix Factorization (NMF). Optimal Transport (OT) previously contributed to this case, either through a metric over histograms (Rolet et al., 2016) or by defining a non-linear way of aggregating atoms (Schmitz et al., 2018) through Wasserstein barycenters (Agueh & Carlier, 2011).

In parallel, different problems in Machine Learning (ML) can be analyzed through a probabilistic view, e.g., generative modeling (Goodfellow et al., 2014) and Domain Adaptation (DA) (Pan & Yang, 2009). For instance, in Multi-Source Domain Adaptation (MSDA), one wants to adapt data from heterogeneous domains or datasets to a new setting. In this case, the celebrated Empirical Risk Minimization (ERM) principle cannot be correctly applied due to the non-i.i.d. character of the data. However, we assume that the domain shifts have regularities that can be learned and leveraged for MSDA. Thus, in this paper, we take a novel approach to MSDA, using *distributional* DiL: we learn a dictionary of empirical distributions. As such, we reconstruct domains using interpolations in the Wasserstein space, also known as Wasserstein barycenters. As we explore in section 3, this offers a principled framework for MSDA.

We take inspiration from the works of Bonneel et al. (2016) and Schmitz et al. (2018) for defining our novel DiL framework. Indeed, these authors consider DiL over histograms, while we propose a DiL over datasets, understood as point clouds, which enables its application to DA. We summarize our contributions as follows,

- **Dictionary Learning.** To the best of our knowledge, we are the first to propose a DiL problem over point clouds.

- **Empirical Distributions Embedding.** As a by-product, we get embeddings of the DiL datasets as their barycentric coordinates w.r.t. the dictionary. We build on this new representation to define a (semi-)metric called Wasserstein Embedding Distance (WED). We explore the WED theoretically (theorems 3.2 and 3.3) and in experiments (section 4.2).

- **Domain Adaptation.** We propose two novel ways for performing MSDA. The first relies on reconstructing labeled samples in the support of the target distribution. The second relies on weighting predictors learned on each atom, thus defining a new classifier that works on the target domain. We offer theoretical justification for both methods (section 3.1). We further explore, in section 4.1.3, general interpolations in the latent space of our dictionary.

**Notation.** We denote as $\Delta_N = \{\mathbf{a} \in \mathbb{R}_+^N : \sum_{i=1}^N a_i = 1\}$ the $N$ probability simplex. We consider $n_P$ i.i.d. samples $\mathbf{X}^{(P)} = \{\mathbf{x}_i^{(P)}\}_{i=1}^{n_P} \in \mathbb{R}^{n_P \times d}$ from an unknown distribution $P$: $\mathbf{x}_i^{(P)} \sim P$.

The samples $\mathbf{X}^{(P)}$ yield an empirical approximation, $\hat{P}$ of $P$ as a sum of delta Diracs, $\hat{P} = \sum_{i=1}^{n_P} p_i \delta_{\mathbf{x}_i^{(P)}}$, with $p_i = 1/n_P$ unless stated otherwise. Similarly, $\hat{Q} = \sum_{i=1}^{n_Q} q_i \delta_{\mathbf{x}_i^{(Q)}}$, with $q_i = 1/n_Q$, is an empirical approximation of a probability distribution $Q$ on $\mathbb{R}^d$. Additionally, each $\mathbf{x}_i^{(P)}$ may have a label $y_i^{(P)} \in \{1, \cdots, n_c\}$ or $\mathbf{Y}_i^{(P)} \in \Delta_{n_c}$ for its one-hot encoding. In this case $\hat{P} = \sum_{i=1}^{n_P} p_i \delta_{(\mathbf{x}_i^{(P)}, y_i^{(P)})}$ is an empirical approximation for the joint $P(X, Y)$.

**Paper Structure.** Section 2 covers brief introductions to OT, DA, and DiL. Section 3 introduces our view on dictionary learning. Section 4 covers experiments in manifold learning of distributions and domain adaptation. Section 5 presents our conclusions.

## 2 PRELIMINARIES

### 2.1 OPTIMAL TRANSPORT

In this section, we focus on the computational treatment of OT (Peyré et al., 2019), which predominantly relies on empirical approximations of distributions. There are two discretization strategies. First, known as *Eulerian discretization*, one seeks to bin $\mathbb{R}^d$ into a fixed grid $\{\mathbf{x}_i^{(P)}\}_{i=1}^{n_P}$ so that $p_i$ corresponds to how many samples are assigned to the $i-$th bin. The second, known as *Lagrangian discretization*, assume $\mathbf{x}_i^{(P)}$ i.i.d. according to $P$. Henceforth, contrary to Bonneel et al. (2016); Schmitz et al. (2018), we use the *Lagrangian discretization*.

For $\hat{P}, \hat{Q}$, in the formulation of Kantorovich (1942) OT seeks a transport plan $\pi \in \mathbb{R}^{n_P \times n_Q}$, where $\pi_{i,j}$ represents how much mass transported from $\mathbf{x}_i^{(P)}$ to $\mathbf{x}_j^{(Q)}$. $\pi$ is required to preserve mass, that is, $\sum_{i=1}^{n_P} \pi_{i,j} = q_j$ and $\sum_{j=1}^{n_Q} \pi_{i,j} = p_i$, or $\pi \in U(\mathbf{p}, \mathbf{q})$, the set of bi-stochastic matrices with marginals $\mathbf{p} \in \Delta_{n_P}, \mathbf{q} \in \Delta_{n_Q}$. Therefore,

$$\pi^\star = \mathrm{OT}(\mathbf{p}, \mathbf{q}, \mathbf{C}) = \underset{\pi \in U(\mathbf{p}, \mathbf{q})}{\mathrm{argmin}} \sum_{i=1}^{n_P} \sum_{j=1}^{n_Q} C_{i,j} \pi_{i,j} = \langle \mathbf{C}, \pi \rangle_F, \tag{1}$$

is the OT problem between $\hat{P}$ and $\hat{Q}$. In Equation 1, $C_{i,j} = c(\mathbf{x}_i^{(P)}, \mathbf{x}_j^{(Q)})$ is called *ground-cost matrix*. When $c$ is a distance, OT defines a distance between distributions (Peyré et al., 2019) called Wasserstein distance, $W_c(\hat{P}, \hat{Q}) = \langle \mathbf{C}, \pi^\star \rangle_F$. For $\mathcal{P} = \{\hat{P}_k\}_{k=1}^K$ and $\boldsymbol{\alpha} \in \Delta_K$, the Wasserstein barycenter (Agueh & Carlier, 2011) is a solution to,

$$B^\star = \mathcal{B}(\boldsymbol{\alpha}; \mathcal{P}) = \inf_B \sum_{k=1}^K \alpha_k W_c(P_k, B). \tag{2}$$

Henceforth we call $\mathcal{B}(\cdot; \mathcal{P})$ *barycentric operator*. For empirical $\hat{P}_k$, estimating $\hat{B}^\star$ is done through the fixed-point iterations of Álvarez-Esteban et al. (2016),

$$(\mathbf{C}_k)_{i,j} = \|\mathbf{x}_i^{(P_k)} - \mathbf{x}_j^{(B)}\|_2^2; \ \pi_k = \mathrm{OT}(\mathbf{p}_k, \mathbf{b}, \mathbf{C}_k); \ \mathbf{X}^{(B)} = \sum_{k=1}^K \alpha_k \mathrm{diag}(\mathbf{b})^{-1} \pi_k^T \mathbf{X}^{(P_k)}, \tag{3}$$

until convergence. In this case $\mathbf{b} \in \Delta_n$ (e.g., $b_j = 1/n$). We further discuss iterations in equation 3 in appendix A.

## 2.2 Empirical Risk Minimization and Domain Adaptation

In traditional ML, the classification problem can be formalized through the ERM principle of Vapnik (1991), which consists in finding $\hat{h}^\star$, which verifies

$$\hat{h}^\star = \underset{h \in \mathcal{H}}{\operatorname{argmin}} \, \hat{\mathcal{R}}_P(h) = \frac{1}{n} \sum_{i=1}^{n} \mathcal{L}(h(\mathbf{x}_i^{(P)}), y_i^{(P)}),$$

where $\mathcal{L}$ is a loss function and $h : \mathbb{R}^d \to \{1, \cdots, n_c\}$ is a classifier, chosen among a family $\mathcal{H}$. Under the i.i.d. hypothesis between training and real application data, the classification error expectation quantified as

$$\mathcal{R}_P(h, h_0) = \mathbb{E}_{\mathbf{x} \sim P}[l(h(\mathbf{x}), h_0(\mathbf{x}))], \tag{4}$$

where $h_0$ is the unknown ground-truth labeling function. Given that enough samples are available $\mathcal{R}_P$ and $\hat{\mathcal{R}}_P$ are close with high probability (Redko et al., 2020, Theorem 1).

Nonetheless, the i.i.d. assumption is restrictive since real-world data may be acquired under different regimes. In this case, train and test data may follow different distributions (Quinonero-Candela et al. (2008)). DA (Kouw & Loog, 2019) is a framework for non-standard cases.

Following Pan & Yang (2009), a domain $\mathcal{D} = (\mathcal{X}, Q(X))$ is a pair of a feature space $\mathcal{X}$ (e.g., $\mathbb{R}^d$) and its distribution $Q(X)$. In DA, one has $\mathcal{D}_S \neq \mathcal{D}_T$ due to different distributions, that is, $Q_S \neq Q_T$. The goal is thus adapting a classifier learned with *labeled data* from $\mathcal{D}_S$, using *unlabeled data* from $\mathcal{D}_T$. When $N_S > 1$ sources are available, $\{\mathcal{D}_{S_\ell}\}_{\ell=1}^{N_S}$ (resp. $Q_{S_\ell}$), one has MSDA. DA can be further divided into shallow and deep DA. In the first case, one leverages feature extractors (e.g., pre-trained convolutional layers). In contrast, in the second, one uses unlabeled target data during training for learning features invariant to distributional shifts.

OT has contributed to DA in various ways. For instance, in the seminal works of Courty et al. (2016), the authors proposed transporting source domain samples using $\hat{\mathbf{X}}^{(Q_S)} = \operatorname{diag}(\mathbf{q}_S)^{-1} \pi \mathbf{X}^{(Q_T)}$. In this sense $\hat{\mathbf{X}}^{(Q_S)} \sim Q_T$, thus their method generates labeled data on the target domain. OT has been applied for MSDA as well. For instance, Montesuma & Mboula (2021a;b) proposes to aggregate $\{Q_{S_\ell}\}_{\ell=1}^{N_S}$ using the Wasserstein barycenter $\hat{B}$, then transporting $\hat{B}$ to $\hat{Q}_T$. Otherwise, Turrisi et al. (2022) estimate domain importance coefficients $\boldsymbol{\alpha}$ for weighting source domain distributions.

## 2.3 Dictionary Learning

DiL is a representation learning technique that tries to express a collection of $N$ vectors $\{\mathbf{x}_\ell\}_{\ell=1}^{N}$, $\mathbf{x}_\ell \in \mathbb{R}^d$ through a set of $K$ atoms $\mathcal{P} = \{\mathbf{p}_k\}_{k=1}^{K}$, $\mathbf{p}_k \in \mathbb{R}^d$ and $N$ weights $\mathcal{A} = \{\boldsymbol{\alpha}_\ell\}_{\ell=1}^{N}$, $\boldsymbol{\alpha}_\ell \in \mathbb{R}^K$. Mathematically, DiL corresponds to,

$$(\mathcal{P}^\star, \mathcal{A}^\star) = \underset{\mathcal{P}, \mathcal{A}}{\operatorname{argmin}} \, \frac{1}{N} \sum_{i=1}^{N} \mathcal{L}(\mathbf{x}_\ell, \mathcal{P}^T \boldsymbol{\alpha}_\ell) + \lambda_A \Omega_A(\mathcal{A}) + \lambda_P \Omega_P(\mathcal{P}), \tag{5}$$

where $\mathcal{L}$ is a suitable loss, whereas $\Omega_A$ and $\Omega_P$ are regularization terms of representations and atoms. OT has previously contributed to DiL, when the vectors are histograms, that is $\mathbf{x}_\ell \in \Delta_d$. In this sense, these contributions assume an *Eulerian* discretization paradigm. For instance Rolet et al. (2016) considered using the Sinkhorn divergence (Cuturi, 2013) as a loss in 5, and Schmitz et al. (2018) proposed substituting $\mathcal{P}^T \boldsymbol{\alpha}_\ell$ for the Wasserstein barycenter.

## 3 Dataset Dictionary Learning

We seek to learn a dictionary over *point clouds*. As such, $\mathcal{Q} = \{\hat{Q}_\ell\}_{\ell=1}^{N}$, where each $\hat{Q}_\ell$ has support $\{\mathbf{x}_i^{(Q_\ell)}\}_{i=1}^{n_{Q_\ell}}$ or $\{(\mathbf{x}_i^{(Q_\ell)}, y_i^{(Q_\ell)})\}_{i=1}^{n_{Q_\ell}}$, corresponding to whether $Q_\ell$ is labeled or not. In this setting, we rewrite equation 5 as,

$$(\mathcal{P}^\star, \mathcal{A}^\star) = \underset{\mathcal{P}, \mathcal{A}}{\operatorname{argmin}} \, L(\mathcal{P}, \mathcal{A}) = \frac{1}{N} \sum_{\ell=1}^{N} W_c(\hat{Q}_\ell, \mathcal{B}(\boldsymbol{\alpha}_\ell; \mathcal{P})), \tag{6}$$

where $\mathcal{P} = \{\hat{P}_k\}_{k=1}^K$ is our set of atoms, and $\mathcal{A} = \{\boldsymbol{\alpha}_\ell\}_{\ell=1}^N$ with $\boldsymbol{\alpha}_\ell \in \Delta_K$, $\alpha_{\ell,k}$ denotes how much $\hat{Q}_\ell$ is composed by $\hat{P}_k$ in the Wasserstein distance sense. Each $\hat{P}_k$ a points cloud parametrized by its support $\mathbf{X}^{(P_k)}$ and labels $\mathbf{Y}^{(P_k)}$ when desired. Hence, we use the fixed-point iterations of Álvarez-Esteban et al. (2016), and differentiate $\mathcal{B}(\boldsymbol{\alpha}; \mathcal{P})$ w.r.t. $\boldsymbol{\alpha}$ and $\mathbf{x}_i^{(P_k)}$ using the Envelope theorem (Bertsekas, 1997). More on this matter is discussed in appendix A.

In addition, optimizing 6 over entire datasets might be untractable due the time and memory complexity of $W_c$, which are, given $n$ samples, $\mathcal{O}(n^3 \log n)$ and $\mathcal{O}(n^2)$, respectively. We thus employ mini-batch OT (Fatras et al., 2021). For $M$ mini-batches of size $n_b \ll n$, this reduces complexity to $\mathcal{O}(Mn_b^3 \log n_b)$ and $\mathcal{O}(n_b^2)$, respectively.

As the $\hat{Q}_\ell$ can be either marginal over features, or feature-label joint distributions in equation 6, the atoms $\hat{P}_k$ are joint features-labels distributions, and the ground-cost must take into account both labels and features. We propose using,

$$C_{i,j} = \|\mathbf{x}_i^{(P)} - \mathbf{x}_j^{(Q)}\|_2^2 \\ + \beta\|\mathbf{Y}_i^{(P)} - \mathbf{Y}_j^{(Q)}\|_2^2, \quad (7)$$

for $\beta = \max_{i,j}\|\mathbf{x}_i^{(P)} - \mathbf{x}_j^{(Q)}\|_2^2$. This ground-cost allows us to determine $\mathbf{X}^{(B)}$ using iterations 3 and to formally justify the label propagation approach of Redko et al. (2019),

$$\mathbf{Y}^{(B)} = \sum_{k=1}^K \alpha_k \mathrm{diag}(\mathbf{b})^{-1}\pi_k^T\mathbf{Y}^{(P_k)}, \quad (8)$$

for labeling the samples in $\mathbf{X}^{(B)}$. Similarly to Bonneel et al. (2016) and Schmitz et al. (2018), we optimize w.r.t $\mathbf{A} = \{a_{\ell,k}\} \in \mathbb{R}^{N \times K}$, then perform a change of variables $\boldsymbol{\alpha}_\ell = \mathrm{softmax}(\mathbf{a}_\ell)$. The overall procedure is shown in algorithm 1, which is implemented using Pytorch (automatic differentiation) of Paszke et al. (2019) and Python Optimal Transport of Flamary et al. (2021).

---

**Algorithm 1** Dataset Dictionary Learning (DaDiL) learning loop.

**Require:** $\mathcal{Q} = \{\hat{Q}_\ell\}_{\ell=1}^N$, number of iterations $N_{iter}$, number of atoms $K$, number of batches $M$, batch size $n_b$

1: Initialize $\mathbf{x}_i^{(P_k)} \sim \mathcal{N}(0, \mathbf{I}_d)$, $\mathbf{a}_\ell \sim \mathcal{N}(0, \mathbf{I}_K)$.
2: **for** $it = 1 \cdots, N_{iter}$ **do**
3:    **for** $batch = 1, \cdots, M$ **do**
4:       **for** $\ell = 1, \cdots, N$ **do**
5:          $\forall k$, sample $\mathbf{X}^{(P_k)} = \{\mathbf{x}_i^{(P_k)}\}_{i=1}^{n_b}$,
6:          sample $\mathbf{X}^{(Q_\ell)} = \{\mathbf{x}_j^{(Q_\ell)}\}_{j=1}^{n_b}$,
7:          change variables $\boldsymbol{\alpha}_\ell = \mathrm{softmax}(\mathbf{a}_\ell)$
8:          calculate $\mathbf{X}^{(B_\ell)} = \mathcal{B}(\boldsymbol{\alpha}_\ell; \mathcal{P})$
9:       **end for**
10:      $L = (1/N)\sum_{\ell=1}^N W_c(\hat{B}_\ell, \hat{Q}_\ell)$
11:      $\forall k, i$, update $\mathbf{x}_i^{(P_k)}$ using $\partial L/\partial \mathbf{x}_i^{(P_k)}$
12:      $\forall \ell$, update $\mathbf{a}_\ell$ using $\partial L/\partial \mathbf{a}_\ell$.
13:    **end for**
14: **end for**
**Ensure:** Dictionary $\mathcal{P}^\star$ and weights $\mathcal{A}^\star$.

---

In the following, we present the derived MSDA approaches.

### 3.1 DOMAIN ADAPTATION

We assume that $N_S > 1$ *labeled source distributions*, $\hat{Q}_{S_1}, \cdots, \hat{Q}_{S_{N_S}}$, are available. The goal is to *improve performance* on an unlabeled *target distribution*, $\hat{Q}_T$. Hence, in eq. 6, $\mathcal{Q} = \{\hat{Q}_{S_\ell}(X,Y)\}_{\ell=1}^{N_S} \cup \{\hat{Q}_T(X)\}$ with $\mathcal{P} = \{\hat{P}_k(X,Y)\}_{k=1}^K$ and $\mathcal{A} = \{\boldsymbol{\alpha}_\ell\}_{\ell=1}^{N_S+1}$. By definition of $\mathcal{Q}$, $\boldsymbol{\alpha}_{N_S+1}$ corresponds to the weights of the target distribution, i.e., $\boldsymbol{\alpha}_T$. We now discuss our two strategies, DaDiL-R and DaDiL-E. An overview is shown in figure 1.

**DaDiL-R.** Our first strategy is based on the reconstruction $\hat{B}_T(X,Y) = \mathcal{B}(\boldsymbol{\alpha}_T; \mathcal{P})$ of $\hat{Q}_T$. Features $\mathbf{X}^{(B_T)}$ and labels $\mathbf{Y}^{(B_T)}$ are generated with equations 3 and 8 respectively. Using (Redko et al., 2017, Theorem 2), we can theoretically bound the risks on these domains using,

$$\mathcal{R}_{Q_T}(h) \le \mathcal{R}_{B_T}(h) + W_1(\hat{Q}_T, \hat{B}_T) + \sqrt{2\frac{\log(\frac{1}{\delta})}{\zeta}}\left(\sqrt{\frac{1}{n_{Q_T}}} + \sqrt{\frac{1}{n}}\right) + \lambda. \quad (9)$$

As such, a classifier trained with data from a distribution $\hat{B}_T$ close to $\hat{Q}_T$ is likely to perform well on $\hat{Q}_T$. The other terms in 9 are related to the sample size and the joint error $\lambda = \mathrm{argmin}_{h \in \mathcal{H}}\mathcal{R}_{Q_T}(h) + \mathcal{R}_{B_T}(h)$. Since $\hat{Q}_T$ is unlabeled, $\lambda$ cannot be controlled directly. Similarly to Redko et al. (2017),

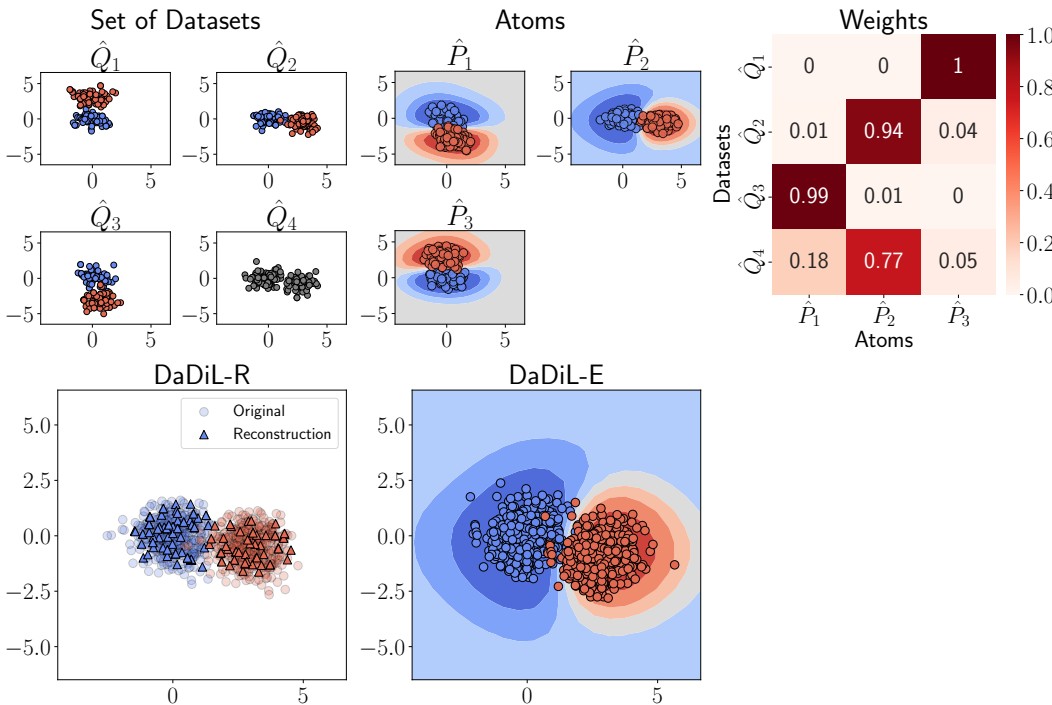

Figure 1: Illustration of DaDiL for MSDA. (Upper left) set of datasets $\mathcal{Q}$. (Upper middle) learned atoms $\hat{P}_k$ and their corresponding classifiers $\hat{h}_k$ (color map). (Upper right) learned weights. (Bottom left) reconstruction $\hat{B}_T$ of $\hat{Q}_T = \hat{Q}_4$. (Bottom middle) Ensembled classifier $\hat{h}_{\boldsymbol{\alpha}_T}$ (color map) and the true labels of $\hat{Q}_T$ (scatter plot).

we hypothesize that the inclusion of labels in the ground-cost and the presence of labels in the other sources help control $\lambda$.

**DaDiL-E.** Our second strategy is based on ensembling. Each atom $\hat{P}_k$ has a *labeled support*, i.e. $\{(\mathbf{x}_i^{(P_k)}, y_i^{(P_k)})\}_{k=1}^K$, for which *we learn a classifier* $\hat{h}_k = \mathrm{argmin}_{h \in \mathcal{H}} \hat{\mathcal{R}}_{P_k}(h)$. To predict target labels, we use the predictor $\hat{h}_{\boldsymbol{\alpha}_T}(\mathbf{x}_j^{(Q_T)}) = \sum_{k=1}^K \alpha_{T,k} \hat{h}_k(\mathbf{x}_j^{(Q_T)})$. We theoretically justify this method as follows,

**Theorem 3.1.** Let $\{\mathbf{X}^{(P_k)}\}_{k=1}^K$ and $\mathbf{X}^{(Q_T)}$ be samples of size $n$ and $n_{Q_T}$ from $P_k$ and $Q_T$. Let $h_k$ be the minimizer of $\mathcal{R}_{P_k}$. Then, for any $d' > d$ and $\zeta > \sqrt{2}$ there exists $N_0$, depending on $d'$ such that for any $\delta > 0$ and $\min(n, n_{Q_T}) > N_0 \max(\delta^{-d'+2}, 1)$ and $\boldsymbol{\alpha} \in \Delta_K$, with probability at least $1 - \delta$, the following holds,

$$\mathcal{R}_{Q_T}(h_{\boldsymbol{\alpha}}) \le \mathcal{R}_{\boldsymbol{\alpha}}(h_{\boldsymbol{\alpha}}) + \sum_{k=1}^K \alpha_k(W_1(\hat{P}_k, \hat{Q}_T) + \lambda_k + c), \tag{10}$$

where $h_{\boldsymbol{\alpha}} = \sum_{k=1}^K \alpha_k h_k$, $\lambda_k = \underset{h \in \mathcal{H}}{\text{minimize }} \mathcal{R}_{P_k}(h) + \mathcal{R}_{Q_T}(h)$,

$$\mathcal{R}_{\boldsymbol{\alpha}}(h) = \sum_{k=1}^K \alpha_k \mathcal{R}_{P_k}(h) \text{ and } c = \sqrt{\frac{2 \log(1/\delta)}{\zeta}} \left( \sqrt{\frac{1}{n}} + \sqrt{\frac{1}{n_{Q_T}}} \right).$$

The bound in Equation 10 is very similar to that of equation 9, but it takes into account how the atom classifiers are weighted to form $h_{\boldsymbol{\alpha}}$. In addition, $\boldsymbol{\alpha}_T$ minimizes the r.h.s. in equation 10. Similarly to DaDiL-R, we assume that using equation 7 helps controlling $\lambda_k$.

In the next subsection, we introduce a proxy for the Wasserstein distance based on the barycentric weights in the learnt dictionary.

## 3.2 A PROXY FOR THE EMPIRICAL WASSERSTEIN DISTANCE

We propose approximating distributions by their projection on the set $\mathcal{M}_{\mathcal{P}} = \{\mathcal{B}(\boldsymbol{\alpha}; \mathcal{P}) : \boldsymbol{\alpha} \in \Delta_K\}$.

Thus, we introduce the following proxy for $W_2$, called WED,

$$\text{WED}(\hat{Q}_1, \hat{Q}_2) = \underset{\pi \in U(\boldsymbol{\alpha}_1, \boldsymbol{\alpha}_2)}{\text{argmin}} \sum_{k_1=1}^{K} \sum_{k_2=1}^{K} \pi_{k_1, k_2} W_c(\hat{P}_{k_1}, \hat{P}_{k_2}), \tag{11}$$

which relies on the notion of Barycentric Coordinate Regression (BCR) (Bonneel et al., 2016, Definition 2), defined as,

$$\boldsymbol{\alpha}^{\star} = \underset{\boldsymbol{\alpha} \in \Delta_K}{\text{argmin}} \, W_c(\mathcal{B}(\boldsymbol{\alpha}; \mathcal{P}), \hat{Q}). \tag{12}$$

**Theorem 3.2.** Let $\hat{Q}_1$ and $\hat{Q}_2$ be 2 empirical distributions, and $\mathcal{P} = \{\hat{P}_k\}_{k=1}^K$ be a set of atoms learned by minimizing equation 6. In this case, the WED is a pseudo-metric over $\mathcal{M}$ (proof in A.3).

Calculating the WED is simpler than $W_c$ since one needs to: (i) compute and store the pairwise $W_c(\hat{P}_{k_1}, \hat{P}_{k_2})$; (ii) solve a small $K \times K$ OT problem. Especially if the distributions $\hat{P}_k$ have much fewer samples than $\hat{Q}_\ell$, calculating the WED is faster than $W_c$. We present a theoretical result bounding the WED by the Wasserstein distance of different terms in our dictionary learning problem,

**Theorem 3.3.** Let $\hat{Q}_1$ and $\hat{Q}_2$ be 2 empirical distributions, and $\mathcal{P} = \{\hat{P}_k\}_{k=1}^K$ be a set of learned atoms. For $\boldsymbol{\alpha}_1$ (resp. $\boldsymbol{\alpha}_2$), the barycentric coordinates (e.g., equation 12) of $\hat{Q}_1$ (resp. $\hat{Q}_2$) and $\hat{B}_1 = \mathcal{B}(\boldsymbol{\alpha}_\ell; \mathcal{P})$ (resp. $\hat{B}_2$) the WED is bounded as follows,

$$\text{WED}(\hat{Q}_1, \hat{Q}_2) \leq W_c(\hat{Q}_1, \hat{Q}_2) + \sum_{k=1}^{K} \alpha_{1,k} W_c(\hat{P}_k, \hat{B}_1) + \alpha_{2,k} W_c(\hat{P}_k, \hat{B}_2)$$
$$+ W_c(\hat{B}_1, \hat{Q}_1) + W_c(\hat{B}_2, \hat{Q}_2).$$

This theorem bounds our proxy by the Wasserstein distance, plus two kinds of terms. The first correspond to the geometry of the learned dictionary, whereas the second correspond to the approximation error (e.g., $W_c(\hat{B}_1, \hat{Q}_1)$), which is explicitly minimized in our algorithm.

## 4 EXPERIMENTS

### 4.1 MULTI-SOURCE DOMAIN ADAPTATION

#### 4.1.1 SHALLOW DOMAIN ADAPTATION

We compare our MSDA methods with five other (shallow) DA algorithms: (i) Subspace Alignment (SA) (Fernando et al., 2013); (ii) Transfer Component Analysis (TCA) (Pan et al., 2010); (iii) Optimal Transport Domain Adaptation (OTDA) (Courty et al., 2016); (iv) Wasserstein Barycenter Transport (WBT) (Montesuma & Mboula, 2021a); (v) Weighted JDOT (WJDOT) (Turrisi et al., 2022). (i) and (ii) are standard algorithms in DA, (iii) is in general the OT baseline, and (iv) and (v) are the State-of-the-Art (SOTA) in shallow MSDA. We further consider the baseline case, which corresponds to training a classifier with the concatenation of source domain data, and evaluating on target domain data.

Table 1: Classification accuracy (in %) of DA methods. Each column represents a target domain for which we report mean ± standard deviation over 5 folds. $^*$ denote results from Montesuma & Mboula (2021a), while † denotes results from Turrisi et al. (2022).

| Method | $A$ | $D$ | $W$ | $C$ | Avg |
|---|---|---|---|---|---|
| Baseline | $92.36 \pm 1.21$ | $98.33 \pm 1.56$ | $93.26 \pm 1.88$ | $86.58 \pm 1.92$ | 92.63 |
| SA | $88.61 \pm 1.72$ | $92.08 \pm 3.82$ | $79.33 \pm 3.67$ | $73.00 \pm 2.31$ | 83.26 |
| TCA$^*$ | $86.83 \pm 4.71$ | $89.32 \pm 1.33$ | $\underline{97.51 \pm 1.18}$ | $80.79 \pm 2.65$ | 88.61 |
| OTDA | $88.26 \pm 1.36$ | $90.41 \pm 3.86$ | $88.09 \pm 3.80$ | $83.02 \pm 1.67$ | 87.44 |
| WJDOT† | $\mathbf{94.23 \pm 0.90}$ | $\mathbf{100.00 \pm 0.00}$ | $89.33 \pm 2.91$ | $85.93 \pm 2.07$ | 92.37 |
| WBT$^*_{reg}$ | $92.74 \pm 0.45$ | $95.87 \pm 1.43$ | $96.57 \pm 1.76$ | $85.01 \pm 0.84$ | 92.55 |
| DaDiL-R | $93.26 \pm 1.53$ | $97.50 \pm 2.43$ | $\mathbf{98.88 \pm 0.71}$ | $\underline{86.65 \pm 1.13}$ | 94.07 |
| DaDiL-E | $\underline{93.89 \pm 1.81}$ | $\underline{98.33 \pm 0.83}$ | $\mathbf{98.88 \pm 0.71}$ | $\mathbf{87.00 \pm 1.42}$ | **94.52** |

We experiment on the Caltech-Office 10 benchmark, which is the intersection of the Caltech 256 dataset of Griffin et al. (2007) and the Office 31 dataset of Saenko et al. (2010). This benchmark

consists of four domains: Amazon (A), dSLR (D), Webcam (W) and Caltech (C). More information on these can be found in appendix B. A summary of our experiments is presented in table 1.

We focus on our analysis w.r.t. the SOTA. Our method improves average performance over WJDOT and WBT, being especially better on Webcam and Caltech domains. W.r.t. WJDOT, Turrisi et al. (2022) show in their experiments that their algorithm chooses one source domain (optimal w.r.t. domain similarity) for the adaptation. This yields the best performance on domains $A$ and $D$. On the other hand, ours combine information learned through the atoms for predicting on the target domain, which proves to be better for domains $W$ and $C$. Note that a similar method to ours, WBT, also performs well on these domains. Nonetheless, as we learn to direct reconstruct the target distribution, we manage to outperform their method.

### 4.1.2 DEEP DOMAIN ADAPTATION

For deep DA, we compare six methods from the SOTA; (i) Domain Adversarial Neural Network (DANN) (Ganin et al., 2016); (ii) Wasserstein Distance Guided Representation Learning (WDGRL) (Shen et al., 2018); (iii) Maximum Classifier Discrepancy (MCD) (Saito et al., 2018); (iv) MOST (Nguyen et al., 2021); (v) Moment Matching for MSDA (M3SDA) (Peng et al., 2019) and (vi) WBT (Montesuma & Mboula, 2021a). MOST and WBT constitute the OT-based MSDA SOTA. In our experiments we consider the Office 31 dataset of Saenko et al. (2010) (see appendix B). As the the backbone, we fine-tune a ResNet-50 (He et al., 2016) on the concatenation of source domain data, then perform adaptation with target domain data. A summary of our experiments is shown in table 2.

Table 2: Classification accuracy (in %) of DA methods on the Office 31 benchmark. Each column represents a target domain for which we report mean $\pm$ standard deviation over 5 folds.

| Method | $A$ | $D$ | $W$ | Avg |
|---|---|---|---|---|
| Baseline | 57.45 | 98.00 | 92.45 | 82.63 |
| DANN | 58.69 | 98.00 | 93.08 | 83.25 |
| WDGRL | 59.75 | 98.00 | 94.34 | 84.03 |
| MCD | 62.23 | 98.00 | 94.34 | 84.85 |
| MOST | 64.01 | 97.00 | 96.23 | 85.74 |
| M3SDA | 50.35 | **100.00** | **98.74** | 83.03 |
| WBT$_{reg}$ | 65.78 | 91.00 | 89.94 | 82.24 |
| DaDiL-R | 72.52 | 95.00 | 98.11 | 88.54 |
| DaDiL-E | **73.58** | 96.00 | 96.23 | **88.60** |

On average, our method is SOTA, primarily due to its outstanding performance on the *Amazon* domain, surpassing other methods by a margin of 7.80%. On average, it presents an increase of 2.70% in accuracy. Nonetheless, the transfer towards the *dSLR* domain is negative. In this domain, deep methods such as Margin Disparity Discrepancy (MDD) and M3SDA perform better. Overall, the main difference between our method and the deep methods in the SOTA is that we cannot update the parameters of convolutional layers, thus decreasing performance on this domain.

### 4.1.3 ADAPTATION USING ATOM INTERPOLATIONS

Besides using the inferred target domain weights $\boldsymbol{\alpha}_T = \boldsymbol{\alpha}_{N_S+1}$ for DaDiL-E and DaDiL-R, our method has the potential to generate *infinitely many domains*, by interpolating atoms using an arbitrary $\boldsymbol{\alpha} \in \Delta_K$, that is, through $\mathcal{M}_P = \{\mathcal{B}(\boldsymbol{\alpha}; \mathcal{P}) : \boldsymbol{\alpha} \in \Delta_K\}$. We thus explore the performance on Caltech-Office 10 and Office 31 benchmarks in terms of $h_{\boldsymbol{\alpha}} = \sum_{k=1}^{K} \alpha_k \hat{h}_k$ and $\hat{B}_{\boldsymbol{\alpha}} = \mathcal{B}(\boldsymbol{\alpha}; \mathcal{P})$, we call these interpolation models. This is shown in figure 2 for $K = 3$, which allows for a nice visualization of the latent space. First, in some cases, the atoms themselves provide good predictors for the target domain (e.g., column 2, rows 2 and 3 in figure 2). This implies that DaDiL *learns distributions* that can discriminate between classes. Second, the reconstruction loss is correlated to the performance of DaDiL-R, as regions with low reconstruction error tend to give better predictions on target. This remark agrees with our theoretical discussion (e.g., eq. 9). Further experiments on this idea can be found in appendix B.2.2.

To give an overview of the performance of interpolations in the latent space we show in figure 3 box plots corresponding to the adaptation performance of DaDiL-E and DaDiL-R on the Caltech-Office 10 benchmark, for each target and arbitrary $\boldsymbol{\alpha}$. For domains $A, C$ and $W$, the choice $\boldsymbol{\alpha}_T$ (dotted red line) is clearly above average. Surprisingly, for $D$, this choice is sub-optimal, indicating that other interpolations in the latent space can be useful for adaptation.

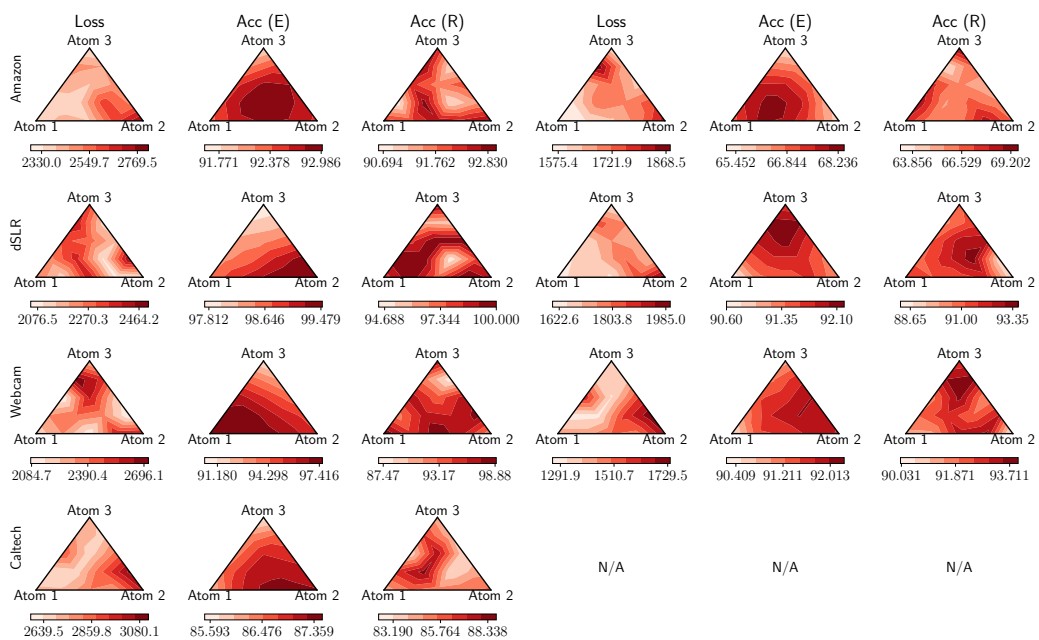

Figure 2: DA performance of interpolations in the latent space of DaDiL. Columns represent reconstruction loss $W_c(\hat{B}_\alpha, \hat{Q}_T)$, and classification accuracy (*Acc*) of DaDiL-E and DaDiL-R, for Caltech-Office 10 (left) and Office 31 (right) benchmarks.

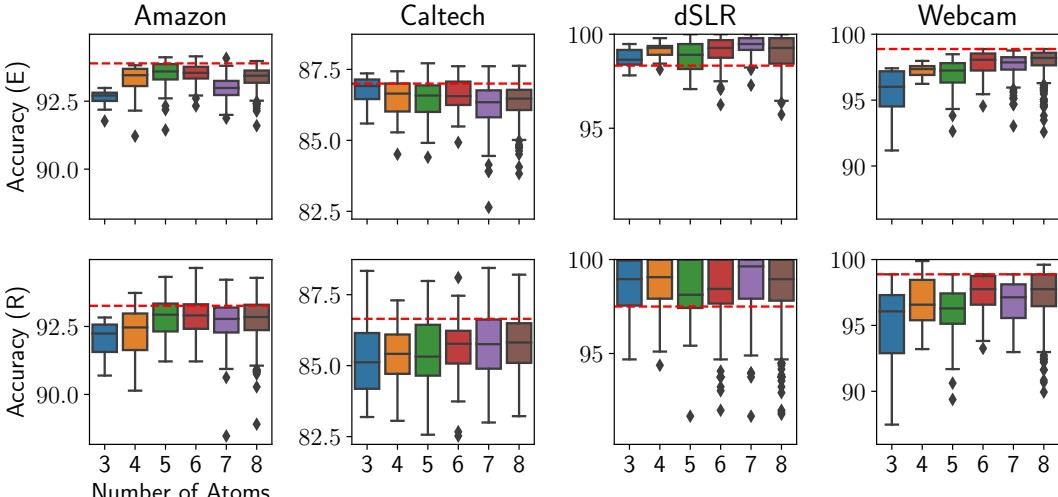

Figure 3: Performance analysis of latent space interpolations on the Caltech-Office 10 benchmark. The red dotted line corresponds to the results reported in Table 1 for DaDiL.

## 4.2 TOY EXAMPLE: LEARNING THE MANIFOLD OF GAUSSIAN DISTRIBUTIONS

In this section we propose to learn a dictionary over the Gaussian distributions. Let $Q_\ell = \mathcal{N}(\mu_\ell, \mathbf{I})$, where $\mu_\ell \in [0, 8]^2 \subset \mathbb{R}^2$ is the mean vector. We discretize $[0, 8]^2$ using 25 uniformly spaced points. For each $Q_\ell$, we sample $n = 512$ points, generating datasets $\hat{Q}_\ell$. In a first moment, we learn a dictionary for $K = 2, \cdots, 6$, which is shown in Figure 4. Except for $K = 2$, all dictionaries provide good reconstructions for the manifold. Furthermore, since $\mu = [\mu_1, \mu_2]$ is a coordinate system for $\mathcal{Q}$, one needs at least three atoms to recover faithful reconstructions.

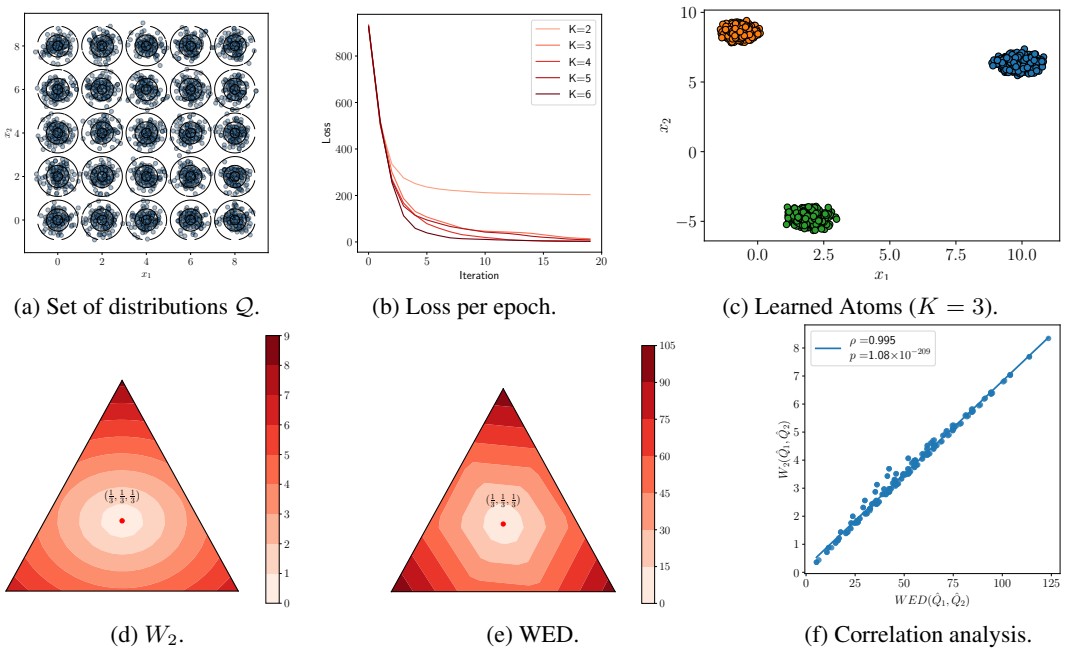

(a) Set of distributions $\mathcal{Q}$.   (b) Loss per epoch.   (c) Learned Atoms ($K = 3$).

(d) $W_2$.   (e) WED.   (f) Correlation analysis.

Figure 4: Overview of DiL on the space of Gaussian distributions.

We focus on $K = 3$ for better visualization. $\alpha_0 = (1/3, 1/3, 1/3)$ and calculate and analyze the $W_2$ and the WED on $\mathcal{M}_P = \{\mathcal{B}(\alpha; \mathcal{P}) : \alpha \in \Delta_3\}$. This is shown in Figures 4 (d) and (e). Since each $Q_\ell \in \mathcal{Q}$ is Gaussian, so is $B$ ((Agueh & Carlier, 2011, Section 6.3)). Therefore $W_2(\hat{B}_0, \hat{B}) = \|\sum_k (\alpha_{0,k} - \alpha_k)\mu_k\|_2^2$, which explains the quadratic contours. Moreover, the WED contours are associated with the positions of the learned atoms and the polytope $U(\alpha_0, \alpha)$, since the ground-cost matrix $C_{k_1,k_2} = W_2(\hat{P}_{k_1}, \hat{P}_{k_2})$ does not depend on $\alpha_0$ or $\alpha$. Indeed, for each $\alpha$, the WED is defined by a $3 \times 3$ OT problem, which has less degrees of freedom (w.r.t. $\alpha \in \Delta_3$) than $W_2$, which is a $512 \times 512$ OT problem. Finally, figure 4 (e) shows that the WED is highly correlated to $W_2$. This is mainly due the fact that our dictionary perfectly reconstructs distributions in $\mathcal{Q}$.

## 5 CONCLUSION

We present a novel probabilistic framework, based on DiL, for learning dictionaries over datasets understood as point clouds, $\mathcal{Q} = \{\hat{Q}_\ell\}_{\ell=1}^N$. We learn a dictionary of labeled points clouds $\mathcal{P} = \{\hat{P}_k\}_{k=1}^K$ and weights $\mathcal{A} = \{\alpha_\ell\}_{\ell=1}^N$, where each distribution $\hat{Q}_\ell$ is expressed as a Wasserstein barycenter of atoms, i.e. $\mathcal{B}(\alpha_\ell; \mathcal{P})$. We propose two novel ways of applying our dictionary for MSDA, by either reconstructing the target domain as the combination of atom distributions, or by ensembling classifiers learned on the atoms $\hat{P}_k$. We show that interpolations in the latent space of our dictionary provide good predictors for the target distribution as well. Finally, we define a pseudo-metric for empirical distributions, based on their barycentric coordinates in the dictionary which is a valuable approximation of the exact Wasserstein distance.

**Limitations and perspectives:** first, in DA, our method relies on the quality of feature extractors or embedding functions. In comparison with deep DA, DaDiL is not able to guide representation learning. We plan to pursue this line of work in future research. Second, similarly to Bonneel et al. (2016), our framework is not able to represent distributions outside $\mathcal{M}_\mathcal{P}$. This may lead to inconsistent reconstructions, which could actually be leveraged for novelty detection.

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

## A    Theoretical Details

### A.1    Differentiation

In this section, we discuss the derivatives of (i) $\hat{P} \mapsto W_c(\hat{P}, \hat{Q})$, (ii) $\alpha \mapsto \mathcal{B}(\alpha; \mathcal{P})$, and (iii) $\mathcal{P} \mapsto \mathcal{B}(\alpha; \mathcal{P})$. Since we parametrize empirical distributions by their support, taking derivatives of distribution-valued functionals (e.g., the Wasserstein distance) is equivalent to differentiating w.r.t. the support of the free distribution (e.g., $\hat{P}$).

Considering the Wasserstein distance, one has two alternatives: (i) using the Sinkhorn algorithm (Cuturi (2013)) for calculating $\pi_\epsilon^\star \approx \pi^\star$, where $\epsilon$ is a parameter that controls the amount of entropic regularization. Since the Sinkhorn algorithm relies on differentiable operations, one can backpropagate through its iterations. (ii) Differentiating through the *min/inf* using the Envelope or Danskin's Theorem Afriat (1971); Bertsekas (1997) at optimality.

The Sinkhorn algorithm has mainly two drawbacks. First, it is unstable numerically for low regularization values, leading to problematic optimization. As remarked by Xie et al. (2020), a possible workaround would be using stabilized versions of the algorithm, but these come with costly exponential/logarithmic operations. The second issue is that the computed OT plans are no longer sparse, which may lead to degenerated distributions (e.g., take the extreme case when $\epsilon \to \infty$). For those reasons, we decide to use exact OT between mini-batches.

A second approach is using the Envelope theorem, which, as advocated by Feydy et al. (2019), can be faster than differentiating through Sinkhorn's iterations. For completeness, we state the Envelope theorem in the following,

**Theorem A.1.** (Envelope Theorem Bertsekas (1997)) Let $Z \subset \mathbb{R}^m$ be a compact set and let $\phi : \mathbb{R}^n \times Z \to \mathbb{R}$ be continuous and such that $\phi(\cdot, \mathbf{z}) : \mathbb{R}^n \to \mathbb{R}$ is convex for each $\mathbf{z} \in Z$. The function $f : \mathbb{R}^n \to \mathbb{R}$ given by,

$$f(\mathbf{x}) = \max_{\mathbf{z} \in Z} \phi(\mathbf{x}, \mathbf{z}),$$

is convex. When $Z(\mathbf{x}) = \{\bar{\mathbf{z}} : \phi(\mathbf{x}, \bar{\mathbf{z}}) = \max_{\mathbf{z}} \phi(\mathbf{x}, \mathbf{z})\}$ consists on an unique point $\bar{\mathbf{z}}$, then

$$\frac{\partial f}{\partial x_i} = \frac{\partial \phi(\mathbf{x}, \bar{\mathbf{z}})}{\partial x_i}. \tag{13}$$

The application of this theorem for OT is straightforward. In this case $Z = U(\mathbf{a}, \mathbf{b}) \subset \mathbb{R}^{m=n_P \times n_Q}$ which is compact. Naturally, the variable $\mathbf{z} = \pi$, whereas $\phi(\mathbf{C}, \pi) = \langle \mathbf{C}, \pi \rangle_F$. This is a linear function of $\mathbf{C}$, hence convex. Finally, the uniqueness of $\pi^\star$ depends on the regularity of the ground-cost. This property is particularly true for the Euclidean distance. In this case, the Wasserstein distance's derivative reads as,

$$W_2(\hat{P}, \hat{Q}) = \sum_{i=1}^n \sum_{j=1}^m \pi_{i,j}^\star \|\mathbf{x}_i^{(P)} - \mathbf{x}_j^{(Q)}\|_2^2 \to \frac{\partial W_2}{\partial \mathbf{X}^{(P)}}(\hat{P}, \hat{Q}) = \frac{2}{n_P} \mathbf{X}^{(P)} - 2(\pi^\star)^T \mathbf{X}^{(Q)},$$

Here, it is essential to note that due to equation 13, to be able to evaluate $\partial W_2/\partial \mathbf{x}_i^{(P)}$, one needs to compute $\pi^\star = \mathrm{OT}(\mathbf{p}, \mathbf{q}, \mathbf{C})$ at optimality. This calculation poses a problem for the Sinkhorn algorithm since the number of iterations needed to converge is inversely proportional to $\epsilon$. Thus, in some cases, it is more efficient to compute exact OT.

Now, we may apply the Envelope theorem to the iterations of Álvarez-Esteban et al. (2016) in equation 3. Let $\pi_k$ be the OT plan between $\mathbf{X}^{(P_k)}$ and $\mathbf{X}^{(B)}$ at convergence. The barycentric operator is thus,

$$\mathbf{X}^{(B)} = \mathcal{B}(\boldsymbol{\alpha}; \mathcal{P}) = \sum_k \alpha_k \pi_k^T \mathbf{X}^{(P_k)},$$

which is linear w.r.t. $\alpha_k$ and $\mathbf{x}_i^{(P_k)}$. The derivatives are, therefore,

$$\frac{\partial \mathbf{X}^{(B)}}{\partial \alpha_k} = \pi_k \mathbf{X}^{(P_k)} \in \mathbb{R}^{n \times d} \text{ and } \frac{\partial \mathbf{x}_j^{(B)}}{\partial \mathbf{x}_i^{(P_k)}} = \alpha_k \mathrm{diag}((\pi_k)_{i,j}) \in \mathbb{R}^{d \times d}.$$

A.2 WASSERSTEIN BARYCENTERS WITH FIXED-POINT ITERATIONS

In previous work (Montesuma & Mboula, 2021a;b), authors have used the iterations of Cuturi & Doucet (2014) for calculating the support $\mathbf{X}^{(B)}$ of Wasserstein barycenters. A similar approach is using the fixed-point approach of Álvarez-Esteban et al. (2016). Let $T_{k,it}$ be the barycentric mapping between $\hat{P}_k$ and $\hat{B}^{(it)}$, that is, $T_{k,it}(\mathbf{x}_j^{(B^{(it)})}) = \frac{1}{b_j} \sum_{i=1}^{n} (\pi_k)_{ij} \mathbf{x}_i^{(P_k)}$, where $\pi_k$ is the OT plan between $\mathbf{X}^{(P_k)}$ and $\mathbf{X}^{(B^{(it)})}$. Moreover, let $T_{it}(\mathbf{x}) = \sum_{k=1}^{K} \alpha_k T_{k,it}(\mathbf{x})$. Álvarez-Esteban et al. (2016) introduce the following mapping,

$$\psi(\hat{P}) = T_{it,\sharp}\hat{P} = \frac{1}{n} \sum_{i=1}^{n} \delta_{T_{it}(\mathbf{x})}.$$

In this setting, Álvarez-Esteban et al. (2016) prove that the Wasserstein barycenter is a fixed-point solution to $\psi$, namely, $\psi(\hat{B}) = \hat{B}$. Their procedure is then equivalent to $\hat{B}^{(it+1)} = \psi(\hat{B}^{(it)})$. This iteration corresponds exactly to those in Equation 3. These are summarized in Algorithm 2.

In section 3, Equation 7 introduced a supervised ground-cost between pairs of samples $(\mathbf{x}_i^{(P)}, y_i^{(P)})$ and $(\mathbf{x}_i^{(Q)}, y_i^{(Q)})$, $C_{i,j} = \|\mathbf{x}_i^{(P)} - \mathbf{x}_j^{(Q)}\|_2^2 + \beta\|\mathbf{Y}_i^{(P)} - \mathbf{Y}_j^{(Q)}\|_2^2$, where $\beta \geq 0$ defines how costly it is to transport samples from different classes. For one-hot encoded vectors $\mathbf{Y} \in \mathbb{R}^{n \times n_c}$, the Euclidean distance is equivalent to the 0-1 loss, $\|\mathbf{Y}_i^{(P)} - \mathbf{Y}_j^{(Q)}\|_2^2 = \delta(y_i^{(P)} - y_j^{(Q)})$. This makes a direct link between our proposed ground-cost, and the semi-supervised regularization proposed by Courty et al. (2016).

In addition, using an Euclidean cost for the labels allows us to interpolate distributions $P(X, Y)$ and $Q(X, Y)$, with respect to features and labels. Indeed, taking derivatives as in the previous section,

$$W_c(\hat{P}, \hat{Q}) = \sum_{i=1}^{n} \sum_{j=1}^{m} \pi_{i,j}^{\star}(\|\mathbf{x}_i^{(P)} - \mathbf{x}_j^{(Q)}\|_2^2 + \beta\|\mathbf{Y}_i^{(P)} - \mathbf{Y}_j^{(Q)}\|_2^2),$$

$$\rightarrow \partial W_c / \partial \mathbf{X}^{(P)}(\hat{P}, \hat{Q}) = \frac{2}{n_P}\mathbf{X}^{(P)} - 2(\pi^{\star})^T \mathbf{X}^{(Q)}, \tag{14}$$

$$\rightarrow \partial W_c / \partial \mathbf{Y}^{(P)}(\hat{P}, \hat{Q}) = \frac{2\beta}{n_P}\mathbf{Y}^{(P)} - 2\beta(\pi^{\star})^T \mathbf{Y}^{(Q)}. \tag{15}$$

Equating the derivatives in 14 and 15 to 0 yields, respectively, the barycentric mapping of Courty et al. (2016) and the label propagation formula of Redko et al. (2019). This allows us to reformulate the fixed-point iterations for joint distributions. Algorithms 2 and 3 show the proposed iterations for barycenters of features and features-labels distributions respectively.

---

**Algorithm 2** Free-Support Wasserstein Barycenter of Unlabeled Distributions

**Require:** $\{\mathbf{X}^{(P_k)}\}_{k=1}^{K}, \boldsymbol{\alpha} \in \Delta_K, \tau > 0, n.$
1: $\mathbf{X}^{(B^{(0)})} = initialize(\{\mathbf{X}^{(P_k)}\}_{k=1}^{K})$
2: **while** $\|\mathbf{X}^{(B^{(it)})} - \mathbf{X}^{(B^{(it-1)})}\|_F \leq \tau$ has not converged **do**
3:     **for** $k = 1, \cdots K$ **do**
4:       $(\mathbf{C}_k)_{i,j} = \|\mathbf{x}_i^{(P_k)} - \mathbf{x}_j^{(B^{(it)})}\|_2^2.$
5:       $\pi_k = \mathrm{OT}(\mathbf{u}_{n_k}, \mathbf{u}_n, \mathbf{C}_k)$
6:     **end for**
7:     $\mathbf{X}^{(B^{(it+1)})} = n \sum_{k=1}^{K} \alpha_k \pi_k^T \mathbf{X}^{(P_k)}$
8: **end while**
**Ensure:** Barycenter support samples $\mathbf{X}^{(B)}$.

**Algorithm 3** Free-Support Wasserstein Barycenter of Labeled Distributions

**Require:** $\{\mathbf{X}^{(P_k)}, \mathbf{Y}^{(P_k)}\}_{k=1}^{K}, \boldsymbol{\alpha} \in \Delta_K, \tau > 0, n, \beta \geq 0.$
1: $\mathbf{X}^{(B^{(0)})}, \mathbf{Y}^{(B^{(0)})} = initialize(\{\mathbf{X}^{(P_k)}, \mathbf{Y}^{(P_k)}\}_{k=1}^{K})$
2: **while** $\|\mathbf{X}^{(B^{(it)})} - \mathbf{X}^{(B^{(it-1)})}\|_F \leq \tau$ has not converged **do**
3:     **for** $k = 1, \cdots K$ **do**
4:       $(\mathbf{C}_k)_{i,j} = \|\mathbf{x}_i^{(P_k)} - \mathbf{x}_j^{(B^{(it)})}\|_2^2 + \beta\|\mathbf{Y}_i^{(P_k)} - \mathbf{Y}_j^{(B^{(it)})}\|_2^2.$
5:       $\pi_k = \mathrm{OT}(\mathbf{u}_{n_k}, \mathbf{u}_n, \mathbf{C}_k)$
6:     **end for**
7:     $\mathbf{X}^{(B^{(it+1)})} = n \sum_{k=1}^{K} \alpha_k \pi_k^T \mathbf{X}^{(P_k)}$
8:     $\mathbf{Y}^{(B^{(it+1)})} = n \sum_{k=1}^{K} \alpha_k \pi_k^T \mathbf{Y}^{(P_k)}$
9: **end while**
**Ensure:** Barycenter support samples $\mathbf{X}^{(B)}$ and labels $\mathbf{Y}^{(B)}$.

---

**Remark 1:** in Algorithms 2 and 3 we use the routine *initialize* to get the initial barycenter support $\mathbf{X}^{(B^{(0)})}, \mathbf{Y}^{(B^{(0)})}$. In our experiments we sub-sample $n$ points from the concatenated matrices $\{\mathbf{X}^{(P_k)}, \mathbf{Y}^{(P_k)}\}_{k=1}^{K}$. Alternatively, one could do as Montesuma & Mboula (2021a;b) and initialize the barycenter support randomly.

### A.3 Approximation Properties of WED

**Proof of Theorem 3.2:** our proof is done in 4 steps: (i) $\text{WED}(\hat{Q}_1, \hat{Q}_2) \geq 0$, $\forall \hat{Q}_1, \hat{Q}_2$, (ii) $\text{WED}(\hat{Q}_1, \hat{Q}_1) = 0$, $\forall \hat{Q}_1$, (iii) $\text{WED}(\hat{Q}_1, \hat{Q}_2) = \text{WED}(\hat{Q}_2, \hat{Q}_1)$ (iv) $\text{WED}(\hat{Q}_1, \hat{Q}_3) \leq \text{WED}(\hat{Q}_1, \hat{Q}_2) + \text{WED}(\hat{Q}_2, \hat{Q}_3)$.

(i) Since $\pi_{k_1, k_2}$ and $W_c(\hat{P}_{k_1}, \hat{P}_{k_2})$ are non-negative, $\forall \hat{Q}_1, \hat{Q}_2$ one has $\text{WED}(\hat{Q}_1, \hat{Q}_2) \geq 0$.

(ii) First, note that $W_c(\hat{Q}_1, \hat{Q}_1) = 0$. Now, since equation 12 is a convex optimization problem, to each $\hat{Q}_1$ one has a single solution $\boldsymbol{\alpha}_1$. Hence, $\mathbf{I}_K \in U(\boldsymbol{\alpha}_1, \boldsymbol{\alpha}_1)$, and $\text{WED}(\hat{Q}_1, \hat{Q}_1) = 0$.

(iii) If $\pi \in U(\boldsymbol{\alpha}_1, \boldsymbol{\alpha}_2)$, then $\pi^T \in U(\boldsymbol{\alpha}_2, \boldsymbol{\alpha}_1)$. Due to commutativity of the summation, $\text{WED}(\hat{Q}_1, \hat{Q}_2) = \text{WED}(\hat{Q}_2, \hat{Q}_1)$.

(iv) Let $\pi^{(1)}, \pi^{(2)}$ and $\pi^{(3)}$ be the solutions to $\text{WED}(\hat{Q}_1, \hat{Q}_3)$, $\text{WED}(\hat{Q}_1, \hat{Q}_2)$ and $\text{WED}(\hat{Q}_2, \hat{Q}_3)$, respectively. In this case,

$$\text{WED}(\hat{Q}_1, \hat{Q}_3) = \sum_{k_1=1}^{K} \sum_{k_2=1}^{K} \pi^{(1)}_{k_1, k_2} W_c(\hat{P}_{k_1}, \hat{P}_{k_2}),$$

$$\leq \sum_{k_1=1}^{K} \sum_{k_2=1}^{K} \pi^{(2)}_{k_1, k_2} W_c(\hat{P}_{k_1}, \hat{P}_{k_2}),$$

since $\pi^{(2)}$ is sub-optimal for the pair $(\hat{Q}_1, \hat{Q}_3)$. Now,

$$\text{WED}(\hat{Q}_1, \hat{Q}_3) \leq \sum_{k_1=1}^{K} \sum_{k_2=1}^{K} \pi^{(2)}_{k_1, k_2} W_c(\hat{P}_{k_1}, \hat{P}_{k_2}),$$

$$\leq \sum_{k_1=1}^{K} \sum_{k_2=1}^{K} (\pi^{(2)}_{k_1, k_2} + \pi^{(3)}_{k_1, k_2}) W_c(\hat{P}_{k_1}, \hat{P}_{k_2})$$

$$= \text{WED}(\hat{Q}_1, \hat{Q}_2) + \text{WED}(\hat{Q}_2, \hat{Q}_3).$$

since $\pi^{(3)}$ is non-negative.

**Proof of Theorem 3.3:** our proof relies on the successive application of the triangle inequality for the Wasserstein distance. Let $\pi \in U(\alpha_1, \alpha_2)$ be the minimizer in equation 11. Then,

$$\text{WED}(\hat{Q}_1, \hat{Q}_2) = \sum_{k_1=1}^{K} \sum_{k_2=1}^{K} \pi_{k_1, k_2} W_c(\hat{P}_{k_1}, \hat{P}_{k_2}),$$

$$\leq \sum_{k_1=1}^{K} \sum_{k_2=1}^{K} \pi_{k_1, k_2} (W_c(\hat{P}_{k_1}, \hat{Q}_2) + W_c(\hat{P}_{k_2}, \hat{Q}_2)),$$

$$\leq \sum_{k_1=1}^{K} \sum_{k_2=1}^{K} \pi_{k_1, k_2} (W_c(\hat{P}_{k_1}, \hat{Q}_1) + W_c(\hat{Q}_1, \hat{Q}_2) + W_c(\hat{P}_{k_2}, \hat{Q}_2)).$$

We now break this sum in 3 parts. First, let us consider $W_2(\hat{P}_{k_1}, \hat{Q}_1)$:

$$\sum_{k_1=1}^{K} \sum_{k_2=1}^{K} \pi_{k_1, k_2} W_c(\hat{P}_{k_1}, \hat{Q}_1) = \sum_{k_1=1}^{K} W_c(\hat{P}_{k_1}, \hat{Q}_1) \left( \sum_{k_2=1}^{K} \pi_{k_1, k_2} \right),$$

$$= \sum_{k_1=1}^{K} \alpha_{1, k_1} W_c(\hat{P}_{k_1}, \hat{Q}_1),$$

$$\leq \sum_{k=1}^{K} \alpha_{1, k} (W_c(\hat{P}_k, \hat{B}_1) + W_c(\hat{B}_1, \hat{Q}_1)),$$

where we used the triangle inequality in the last line. As consequence,

$$\sum_{k_1=1}^{K}\sum_{k_2=1}^{K}\pi_{k_1,k_2}W_c(\hat{P}_{k_1},\hat{Q}_1) \leq W_c(\hat{B}_1,\hat{Q}_1) + \sum_{k=1}^{K}\alpha_{1,k}W_c(\hat{P}_k,\hat{B}_1).$$

The exact same process can be repeated for the term involving $W_2(\hat{P}_{k_2},\hat{Q}_2)$, that is,

$$\sum_{k_1=1}^{K}\sum_{k_2=1}^{K}\pi_{k_1,k_2}W_c(\hat{P}_{k_2},\hat{Q}_2) \leq W_c(\hat{B}_2,\hat{Q}_2) + \sum_{k=1}^{K}\alpha_{1,k}W_c(\hat{P}_k,\hat{B}_2).$$

The final term follows from the fact that $W_2(\hat{Q}_1,\hat{Q}_2)$ is independent of both $k_1$ nor $k_2$. Therefore,

$$\sum_{k_1=1}^{K}\sum_{k_2=1}^{K}\pi_{k_1,k_2}W_c(\hat{Q}_1,\hat{Q}_2) = W_c(\hat{Q}_1,\hat{Q}_2)\bigg(\sum_{k_1=1}^{K}\sum_{k_2=1}^{K}\pi_{k_1,k_2}\bigg) = W_c(\hat{Q}_1,\hat{Q}_2).$$

## A.4 THEORETICAL BOUNDS FOR DOMAIN ADAPTATION

In what follows, we consider the theoretical results of Ben-David et al. (2010) and Redko et al. (2017) for giving the theoretical guarantees of both DaDiL-E and DaDiL-R. For completeness, we re-state Lemma 1 and Theorem 1 of Redko et al. (2017), and prove Theorem 3.1.

**Lemma A.1.** (Due to Redko et al. (2017)) Let $P$ and $Q$ be two probability distributions over $\mathbb{R}^d$. Assume that the cost function $c(\mathbf{x}^{(P)}, \mathbf{x}^{(Q)}) = \|\phi(\mathbf{x}^{(P)}) - \phi(\mathbf{x}^{(Q)})\|_{\mathcal{H}_k}$, where $\mathcal{H}_k$ is a reproducing kernel Hilbert space equipped with kernel $\Phi : \mathbb{R}^d \times \mathbb{R}^d \to \mathbb{R}$ induced by $\phi : \mathbb{R}^d \to \mathcal{H}_k$ and $\Phi(\mathbf{x}^{(P)}, \mathbf{x}^{(Q)}) = \langle\phi(\mathbf{x}^{(P)}), \phi(\mathbf{x}^{(Q)})\rangle_{\mathcal{H}_\Phi}$. Assume that the loss function $l_{h,h_0} : \mathbf{x} \mapsto l(h(\mathbf{x}), h_0(\mathbf{x}))$ is convex, symmetric, bounded, obeys the triangular inequality and has the parametric form $|h(\mathbf{x}) - h_0(\mathbf{x})|^q$ for some $q > 0$. Assume that the kernel $\Phi \in \mathcal{H}_\Phi$ is square-root integrable w.r.t. both $P$ and $Q$ and $0 \leq \Phi(\mathbf{x}^{(P)}, \mathbf{x}^{(Q)}) \leq M, \forall \mathbf{x}^{(P)}, \mathbf{x}^{(Q)} \in \mathbb{R}^d$. Then the following holds,

$$\mathcal{R}_Q(h, h') \leq \mathcal{R}_P(h, h') + W_1(P, Q). \tag{16}$$

Lemma A.1 bounds the risk (recall equation 4) of $h$ with respect to $h'$ under distribution $Q$ by the risk under distribution $P$, plus a term depending on the distance between those 2 distributions. Intuitively, if $P$ is close to $Q$, its samples are similar and thus the risk are similar. We are now interested in acquiring bounds for the empirical risks $\hat{\mathcal{R}}_P$ and $\hat{\mathcal{R}}_Q$ in terms of $W_1(\hat{P}, \hat{Q})$, which are quantities we can estimate from data. We start by stating Theorem 1.1 of Bolley et al. (2007),

**Lemma A.2.** (Due to Bolley et al. (2007)) Let $P$ be a probability distribution over $\mathbb{R}^d$, so that for some $\alpha > 0$ we have that $\int_{\mathbb{R}^d} e^{\alpha\|\mathbf{x}\|^2}dP < \infty$ and $\hat{P}$ be its associated empirical approximation with support $\{\mathbf{x}_i^{(P)}\}_{i=1}^n$ drawn independently from $P$. Then, for any $d' > d$ and $\zeta < \sqrt{2}$ there is a constant $n_0$ depending on $d'$ and some square exponential moment of $P$ such that for any $\epsilon > 0$ and $n \geq n_0\max(\epsilon^{-(d'+2)}, 1)$,

$$\mathbb{P}[W_1(\hat{P}, P) > \epsilon] \leq \exp\bigg(-\frac{\zeta}{2}n\epsilon^2\bigg),$$

where $d'$ and $\zeta'$ can be calculated explicitly.

Lemma A.2 states the conditions for which $\hat{P}$ and $P$ are close in the sense of Wasserstein. This last bound is on the form $\mathbb{P}[quantity > \epsilon] < \delta$, that is, with high probability $quantity \leq \epsilon$. These types of bounds are ubiquitous in the theoretical analysis of learning algorithms. We can express $\epsilon$ explicitly in terms of $\delta$,

$$\epsilon = \sqrt{\frac{2}{n\zeta}\log(\frac{1}{\delta})}, \tag{17}$$

which will be useful in the following discussion. These results allowed Redko et al. (2017) to provide theoretical guarantees for the OTDA framework of Courty et al. (2016),

**Theorem A.2.** (Due to Redko et al. (2017)) Under the assumptions of Lemma A.1, let $\mathbf{X}^{(P)}$ and $\mathbf{X}^{(Q)}$ be 2 samples of size $n_P$ and $n_Q$, drawn i.i.d. from $P$ and $Q$. Let $\hat{P}$ and $\hat{Q}$ be the respective empirical approximations. Then for any $d' > d$ and $\zeta < \sqrt{2}$ there exists some constant $N_0$, depending on $d'$ such that for any $\delta > 0$ and $\min(n_P, n_Q) \geq N_0 \max(\delta^{-d'+2}, 1)$ with probability at least $1 - \delta$ for all $h \in \mathcal{H}$, then,

$$\mathcal{R}_Q(h) \leq \mathcal{R}_P(h) + W_1(\hat{P}, \hat{Q}) + \sqrt{2^{\log(1/\delta)/\zeta}}\left(\sqrt{1/n_P} + \sqrt{1/n_Q}\right) + \lambda.$$

This last theorem effectively states that, besides constant terms that depend on the number of samples $n_P$ and $n_Q$, there are 2 factors that determine whether the risk under $Q$ is similar to that under $P$: (i) the distance in distribution between $\hat{P}$ and $\hat{Q}$, (ii) how well can a classifier in $\mathcal{H}$ work on both domains. In practice, what OTDA does is minimizing $W_1(\hat{P}, \hat{Q})$ while keeping $\lambda$ constant (e.g. by enforcing class-sparsity).

Now, let us discuss how these concepts apply to DaDiL-R. Note that our strategy consists on learning with samples obtained by the barycentric distribution $\hat{B}_T = \mathcal{B}(\boldsymbol{\alpha}_T; \mathcal{P})$, since it approximates the target distribution $\hat{Q}_T$. In this sense,

$$\mathcal{R}_{Q_T}(h) \leq \mathcal{R}_{B_T}(h) + W_1(\hat{Q}_T, \hat{B}_T) + \sqrt{2^{\log(1/\delta)/\zeta}}\left(\sqrt{1/n_T} + \sqrt{1/n}\right) + \lambda.$$

In this last equation, the term $W_1(\hat{Q}_T, \hat{B}_T)$ is the reconstruction error for the target domain, which we directly minimize in our objective function (equation 6). The remaining term is $\lambda$, which we cannot directly optimise since target domain labels are not available during training. Nonetheless our algorithm leverages label information from source domains. If we suppose that their class structure is similar to that of the target domain, the regularization term 7 ensures that $\lambda$ remains bounded, since transport plans will not mix classes.

We now focus on Theorem 3.1, which presents a theoretical guarantee for DaDiL-E.

**Proof of Theorem 3.1** This proof relies on the triangle inequality for the risk. Let $h_{T,k}^{\star} = \arg\min_{h \in \mathcal{H}} R_{Q_T}(h) + R_{P_k}(h)$ and $h_{\boldsymbol{\alpha}} = \sum_k \alpha_k R_{P_k}(h_k^{\star})$ for $h_k^{\star} = \arg\min_{h \in \mathcal{H}} R_{P_k}(h)$. Then,

$$
\begin{aligned}
\mathcal{R}_{Q_T}(h_{\boldsymbol{\alpha}}) &\leq \mathcal{R}_{Q_T}(h_{T,k}^{\star}) + \mathcal{R}_{Q_T}(h_{T,k}^{\star}, h_{\boldsymbol{\alpha}}), \\
&= (\mathcal{R}_{Q_T}(h_{T,k}^{\star}, h_{\boldsymbol{\alpha}}) - \mathcal{R}_{P_k}(h_{T,k}^{\star}, h_{\boldsymbol{\alpha}})) + (\mathcal{R}_{Q_T}(h_{T,k}^{\star}) + \mathcal{R}_{P_k}(h_{T,k}^{\star}, h_{\boldsymbol{\alpha}})), \\
&\leq \underbrace{(\mathcal{R}_{Q_T}(h_k^{\star}, h_{\boldsymbol{\alpha}}) - \mathcal{R}_{P_k}(h_k^{\star}, h_{\boldsymbol{\alpha}}))}_{\leq W_1(P_k, Q_T) \text{ (Lemma A.1)}} + \underbrace{(\mathcal{R}_{Q_T}(h_{T,k}^{\star}) + \mathcal{R}_{P_k}(h_{T,k}^{\star}))}_{=\lambda_k} + \mathcal{R}_{P_k}(h_{\boldsymbol{\alpha}}), \\
&\leq \mathcal{R}_{P_k}(h_{\boldsymbol{\alpha}}) + W_1(P_k, Q_T) + \lambda_k.
\end{aligned}
$$

Summing over $k$, weighted by $\alpha$,

$$
\begin{aligned}
\mathcal{R}_{Q_T}(h_{\boldsymbol{\alpha}}) = \sum_k \alpha_k \mathcal{R}_{Q_T}(h_{\boldsymbol{\alpha}}) &\leq \sum_k \alpha_k R_{P_k}(h_{\boldsymbol{\alpha}}) + \sum_k \alpha_k (W_1(P_k, Q_T) + \lambda_k), \\
&= R_{\alpha}(h_{\boldsymbol{\alpha}}) + \sum_k \alpha_k (W_1(P_k, Q_T) + \lambda_k), \\
&\leq R_{\alpha}(h_{\boldsymbol{\alpha}}) + \sum_k \alpha_k (W_1(\hat{P}_k, \hat{Q}_T) + \lambda_k + c_1),
\end{aligned}
$$

This theorem is similar to previous theoretical guarantees for MSDA, such as those of Ben-David et al. (2010), Redko et al. (2017), and $\text{WBT}_{reg}$ (Montesuma & Mboula (2021a)). In particular, DaDiL-E uses $\boldsymbol{\alpha}_T$ for weighting $h_k^{\star}$. Furthermore this choice of $\boldsymbol{\alpha}$ minimizes the r.h.s. of the last equation.

# B  TECHNICAL DETAILS

## B.1  DATA PREPARATION

**Caltech-Office 10:** we use this dataset for shallow DA. As follows, we use the experimental setting of Montesuma & Mboula (2021a), namely, the 5 fold cross-validation partitions and the features (DeCAF 7th layer activations).

**Office 31:** we use this dataset for deep DA. To that end, we download the raw images from its public repository[1]. We train the feature extractor ourselves, which consists on a ResNet50 (He et al. (2016)). The preprocessing steps taken are: (i) resizing each image to $(224, 224, 3)$, and (ii) applying `tf.keras.applications.resnet50.preprocess_input` function. We initialize its parameters with the weights trained on ImageNet, then fine-tune for 51 epochs on each combination of source domains (e.g., $A, D$). Fine-tuning was performed using the ADAPT library of de Mathelin et al. (2021). We understand each epoch as a full pass through the entire dataset. We then use the fine-tuned network to extract features (vectors $\mathbf{x} \in \mathbb{R}^{2048}$) from the target domain only.

**Remark 2:** we use a ResNet backbone since we were not able to reproduce previous results on the standard AlexNet backbone (e.g., Nguyen et al. (2021)).

**Remark 3:** we give the specifics for the Caltech-Office and Office 31 datasets in Table 3. As follows, we train our algorithms with all source domain data (train and test), and train target domain data. For instance, when training a specific algorithm on Office 31 dataset with the setting $A, D \to W$, at training time the algorithm has 2817 labeled samples from $A$, 498 labeled samples from $D$ and 636 unlabeled samples from $W$ available. For evaluation, we use the target test set. In the context of our example, 159 unlabeled samples from $W$.

Table 3: Details about the datasets considered for domain adaptation.

| Dataset | # Classes | Domain | # Training Samples | # Test Samples |
|---------|-----------|--------|--------------------|----------------|
| Caltech-Office 10 | 10 | Amazon | 748 | 210 |
| | | dSLR | 108 | 49 |
| | | Webcam | 224 | 71 |
| | | Caltech | 956 | 224 |
| Office 31 | 31 | Amazon | 2253 | 564 |
| | | dSLR | 398 | 100 |
| | | Webcam | 636 | 159 |

**Artworks dataset:** we select a sub-set of 50 artworks from the Best Artworks of All Time[2], from authors Monet, Delacroix, Magritte, Caravaggio and Van Gogh, with 10 images from each. The artworks were selected so that they group around 3 major color palettes (red, blue and somber).

## B.2  ADDITIONAL EXPERIMENTS

### B.2.1  TOY EXAMPLE: DICTIONARIES OVER COLOR PALETTES

One of the standard applications of OT is the analysis of color histograms (e.g., Ferradans et al. (2014)). This analysis is done by considering RGB images $I_\ell \in \mathbb{R}^{h \times w \times 3}$ as *point clouds* of $n = hw$ samples in $\mathbb{R}^3$. Thus, to each $I_\ell$, one has $\hat{Q}_\ell$ with support $\mathbf{X}^{(Q_\ell)} \in \mathbb{R}^{hw \times 3}$. In this experiment, we want to learn a dictionary over $\mathcal{Q} = \{\hat{Q}_\ell\}_{\ell=1}^N$, $N = 50$ artworks of Monet, Delacroix, Magritte, Caravaggio, and Van Gogh, based on their color histograms. We further investigate the geometry induced by the latent codes $\mathcal{A} \in (\Delta_K)^N$ and the WED compared to the Wasserstein distance. Here we discuss our results for $K = 3$, using $N_{tr} = 35$ images to train our dictionary, leaving the rest $N_{ts} = 15$ for test. We can map these images into the latent space using equation 12. We show the result of DiL in figure 5.

---

[1]`https://faculty.cc.gatech.edu/~judy/domainadapt/`
[2]`https://www.kaggle.com/datasets/ikarus777/best-artworks-of-all-time`

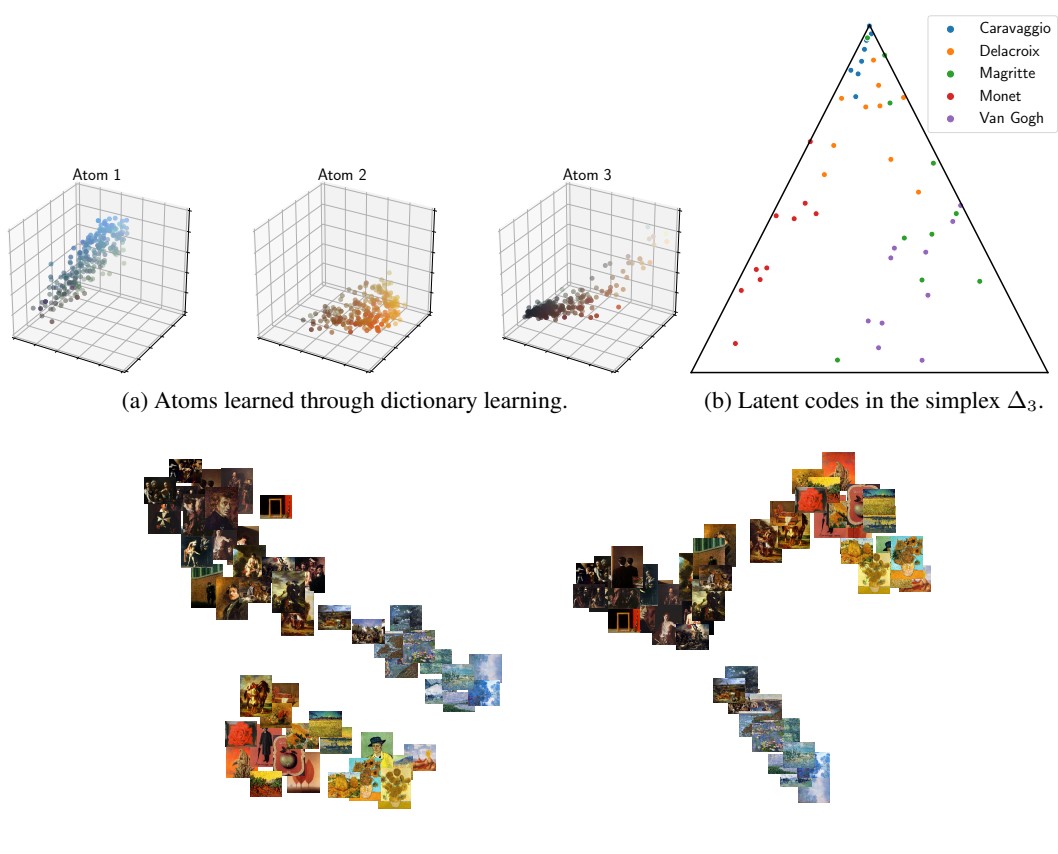

(a) Atoms learned through dictionary learning.

(b) Latent codes in the simplex $\Delta_3$.

(c) Wasserstein distance embeddings

(d) WED embeddings.

Figure 5: Summary of DaDiL for palette learning. Our algorithm learns palettes that reflect different palette clusters (a). These allow for an useful visualization on the latent space (b). The latent codes can be used to approximate the Wasserstein geometry (c) through the proposed WED (d).

We compare the Wasserstein distance with DaDiL and the WED in (i) running time; (ii) their induced geometry. For large-scale images, calculating $W_2(\hat{Q}_{\ell_1}, \hat{Q}_{\ell_2})$ is unfeasible due to the number of pixels. We thus down-sample the support matrices by selecting $n = 2048$ points randomly, hence $\mathbf{X}^{(Q_\ell)} \in \mathbb{R}^{2048 \times 3}$. This step is not necessary for DaDiL since we perform optimization in mini-batches. Concerning running time, the 1225 pairwise $W_2$ distances took approximately 26 minutes[3]. For DaDiL, training takes an overall of 15 minutes. We remark that, for new distributions, we only need to solve 12 over $N_{ts} \times K$ variables. This is usually faster than calculating pairwise distances.

In Figure 5 (a) we show the learned atoms, which correspond to the predominant colors in the training images. Figure 5 (b) shows the latent codes. The embeddings lead to an intuitive description of images, as they express each image as a combination of blue, yellow, and gray palettes. In Figure 5 (c) and (d), we compare the Wasserstein and WED embeddings using t-distributed Stochastic Neighbor Embeddings (t-SNE) (Van der Maaten & Hinton (2008)), which shows that we are able to capture the Wasserstein geometry faithfully.

### B.2.2 SENSITIVITY ANALYSIS

We study how the number of components $K$, the batch size $b$, and the number of samples in the atoms support $n$ influence the performance of DiL. Our results are summarized in Figures 6 and 7. We want to verify whether the DiL loss $L(\mathcal{P}, \mathcal{A})$ is a good predictor for the test score.

---

[3]experiments were performed on a Intel Xeon CPU 2.20 GHz with 12GB of RAM

On one hand, for the Caltech-Office dataset, Figure 6 shows that increasing the number of samples and batch sizes globally leads to a better approximation of the distributions manifold, as $L(\mathcal{P}, \mathcal{A})$ shrinks. This, however, is not necessarily true for the test score, as $b = 128$ and $n = 2048$ leads to better adaptation results. Figures 6 (d) and (e) shows that the DiL is a good proxy for DaDiL-E and DaDiL-R, as it is strongly correlated with the test score. We further remark that $L(\mathcal{P}, \mathcal{A})$ is more correlated with the test score of DaDiL-R, since this method directly relies on the goodness of reconstructions.

On the other hand, for the Office 31 dataset, Figure 7 (a) shows that the dependence on these parameters is much more complex, showing that in general more samples or components do not lead to better approximations. Moreover, Figures 7 (d) and (e) show that $L(\mathcal{P}, \mathcal{A})$ is not correlated with neither DaDiL-R nor DaDiL-E. We believe that since this dataset encompasses more classes than the Caltech-Office, one needs more atoms to model the particularities of each domain.

We now focus our attention on t-SNE embeddings of reconstructions $\hat{B}_T$. Figures 8 and 9 shows the embeddings for the Caltech-Office and Office 31 datasets. For this first dataset, our reconstructions are correctly aligned with the dataset's classes, indicating that the atoms have correctly captured the characteristics of the domains. For the second dataset, as the number of classes is large, we only show a comparison between original and reconstructed samples.

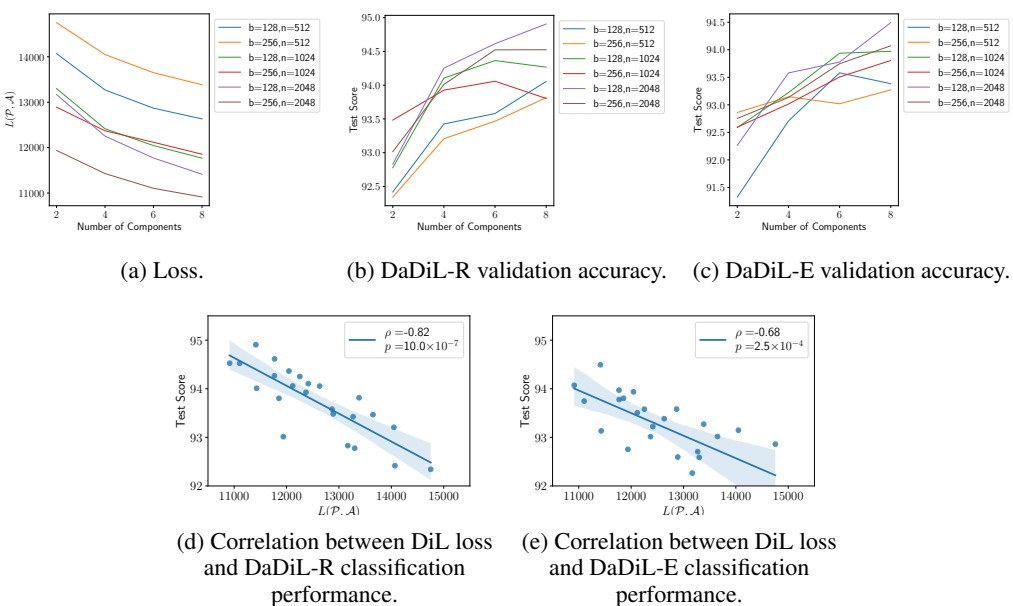

Figure 6: Study of how the batch size, number of samples and number of components impact DiL, DaDiL-R and E performances on Caltech-Office 10 dataset.

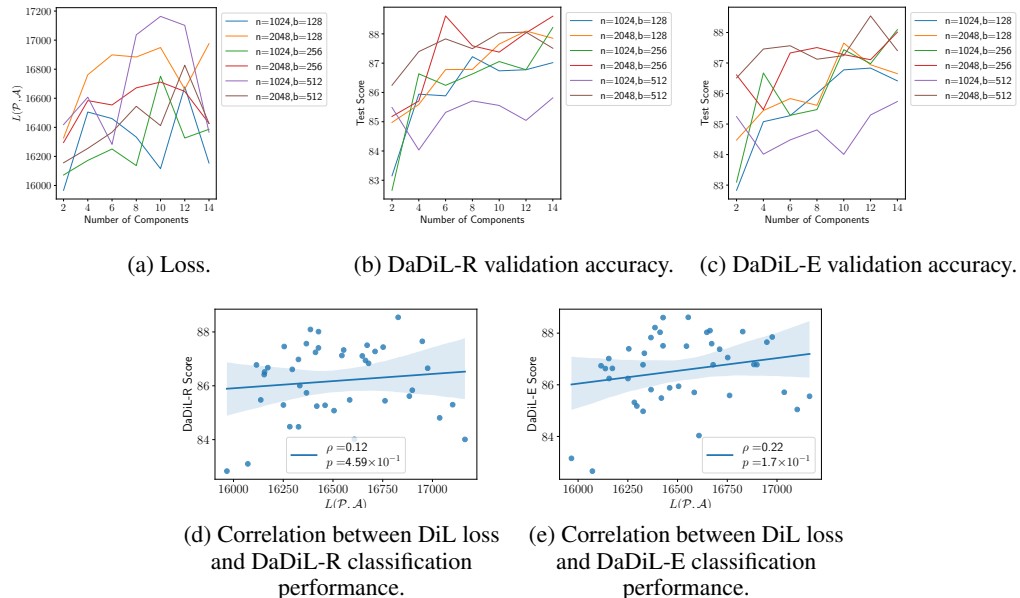

Figure 7: Study of how the batch size, number of samples and number of components impact DiL, DaDiL-R and E performances on Office 31 dataset.

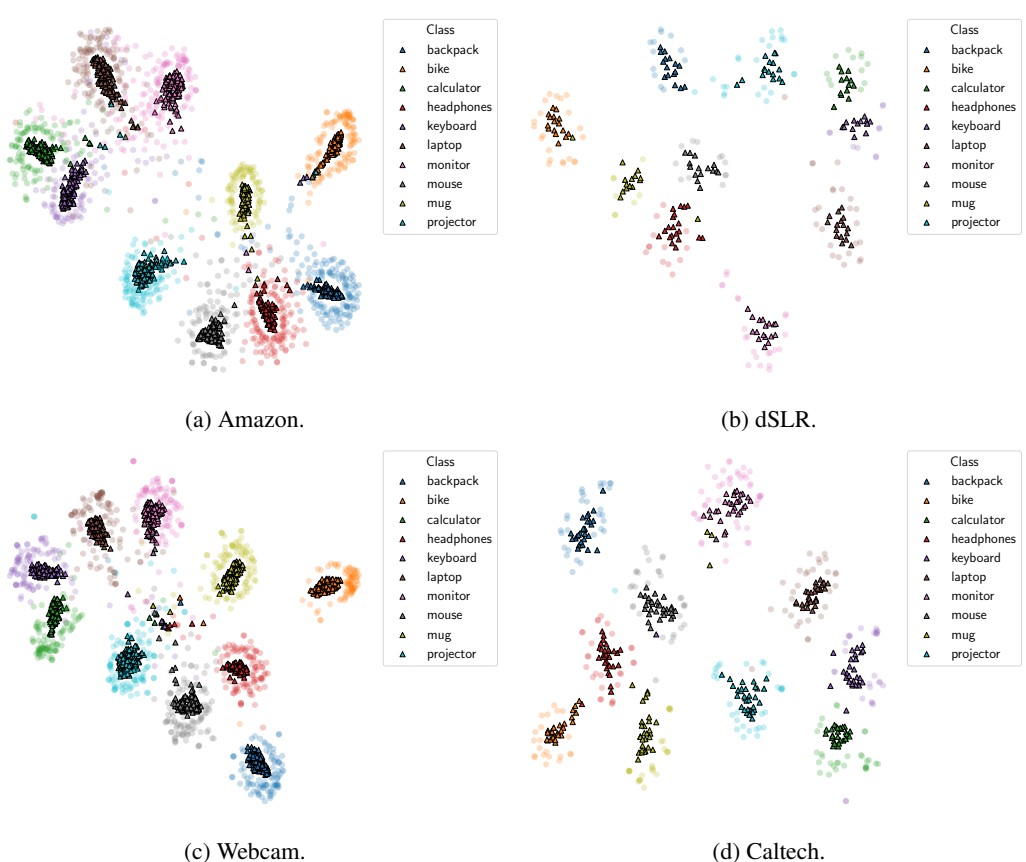

(a) Amazon.

(b) dSLR.

(c) Webcam.

(d) Caltech.

Figure 8: t-SNE visualization of DaDiL-R reconstructions of each target domain. Semi-transparent circles correspond to points in the target domain, whereas triangles correspond to points in $\hat{B}_T$ support.

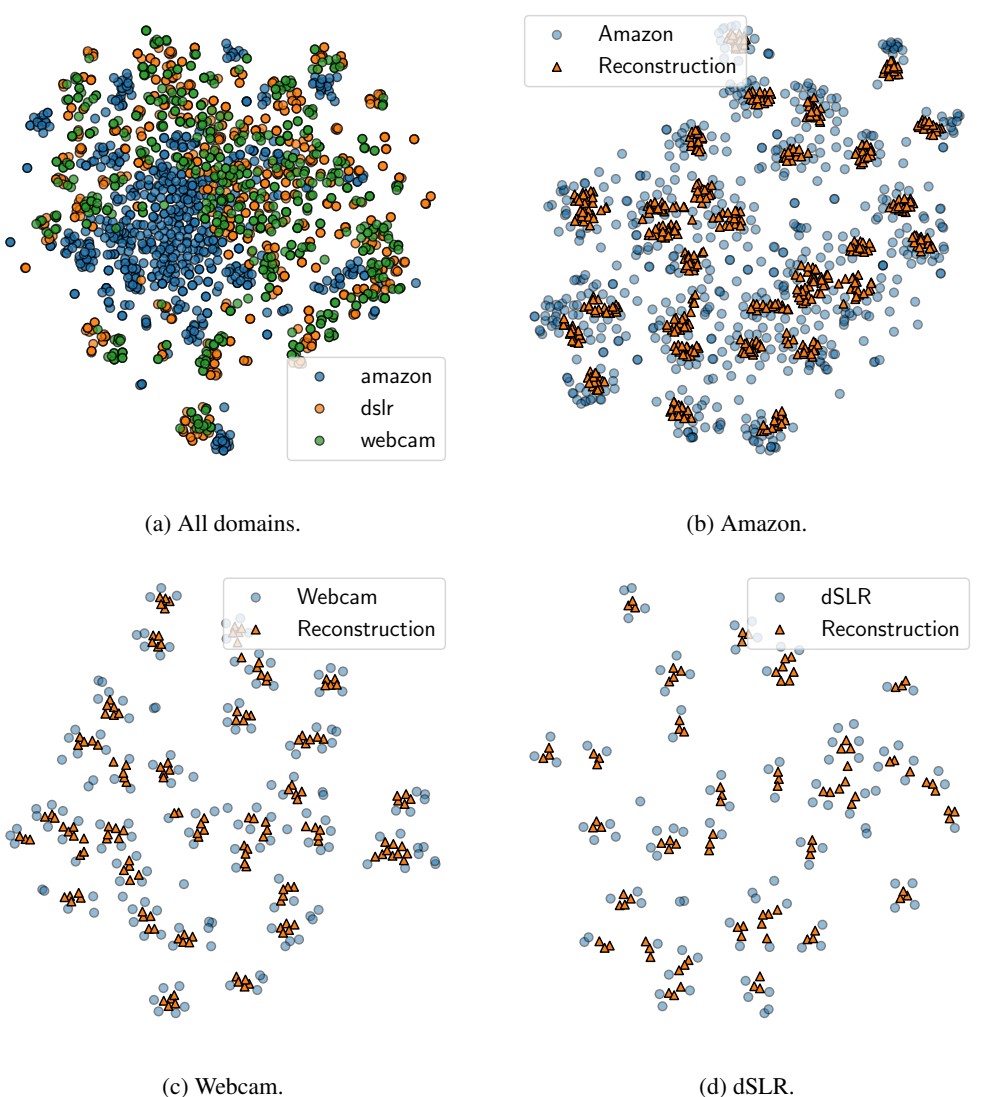

(a) All domains.

(b) Amazon.

(c) Webcam.

(d) dSLR.

Figure 9: t-SNE visualization of DaDiL-R reconstructions of each domain in the Office 31 dataset. Semi-transparent circles correspond to points in the target domain, whereas triangles correspond to points in $\hat{B}_T$ support.

