# OpenReview forum: "Learning Dictionaries over Datasets through Wasserstein Barycenters"
_ICLR.cc/2023/Conference — Submitted to ICLR 2023_

### Official Review · Reviewer_fvrm · 2022-10-26

**Confidence:** 2
**Correctness:** 3
**Technical Novelty And Significance:** 3
**Empirical Novelty And Significance:** 3
**Recommendation:** 5

**Clarity, Quality, Novelty And Reproducibility:**

The paper is clear on the technical aspects and result presentation. It is a novel method to provide domain-adapted dictionaries. Although the method builds upon previously OT-based methods, it has novel contributions and theoretical analysis to the embedding distance between the two datasets based on the learned dictionaries.

**Strength And Weaknesses:**

Strengths: The paper proposes a novel OT-based framework to enable domain adaptation. The approach has advantages over prior works. I recommend its acceptance.

Weaknesses: The paper lacks clear motivation in the abstract and introduction. The organization of the paper makes it hard to distinguish between what is prior work and their work. For example, section 2 says preliminaries, but the authors mention "we propose ..." several times within this section. Please clarify.

Here are some questions that answering may improve the paper and are key to my review:

- How this work differs from the two prior works mentioned in the contribution paragraph?
- The paper lacks a clear motivation. Abstract has no information on what problem this framework may solve and their motivation for pursuing such a model. Why someone should use their method? For example, based on the preliminaries subsection 2.2, is their model trying to address iid assumption made by the majority of the models that may not be valid on a certain application or during the test? This comment of mine is mainly to improve the clarity of the paper.
- Are WJDOT and WBT SOTA? or they are OT-based SOTA? Please clarify.
- What is the complexity of the proposed method compared to other OT methods? In addition, how about a speed comparison?
- Please include C in Table 1 (right). Why is it not included?
- How is the performance compared to autoencoders for Figure 7?
- The ensemble method is not very clear on how it is used for reconstruction. Please elaborate.

Here are some terms that may improve the paper:

- Is there a constraint on the dictionary in optimization formulation (1)?
- I like the toy example. To highlight the advantages of the proposed method, I suggest including a baseline of form OT and dictionary learning in Figure 5.
- What are the dictionaries in Figure 5 left? Possible to visualize them on the same figure?


Minor points:

- I suggest removing the red box around the words.
- There is an extra () for year of the citations. It should be (name, year). You can use \citep instead of \citet.
- The third line after (1) is not a complete sentence. Please rephrase.
- The second line, Section 3, please cite previous works.
- Is the method in Figure 3b DaDiL-E and R?

**Summary Of The Paper:**

The paper proposes an optimal transport-based method to learn dictionaries that generalize over datasets (this is a domain adaptation problem). They propose two approaches: one to reconstruct the samples based on the learned barycenters, and another based on an ensemble of classifiers that are learned based on the dictionary atoms. They provide a toy example of how the method is able to learn a set of dictionaries based on a group of Gaussian-distributed data. They apply their method to learning dictionaries across several datasets and evaluate domain adaptation. They show the outperformance of their method in multi-source domain adaptation and how it performs better than the baselines in both shallow and deep feature generators. At last, they visualize the embedding of the method and how it better encodes similar data close to one another compared to latent based on Wasserstein distance.

**Summary Of The Review:**

The paper proposes an OT-based approach to learning dictionaries across datasets that offer better domain adaptation compared to the baselines. The experimental results are detailed and nicely done. My reservations are coming from clarification on how their method differs from other OT frameworks and the inclusion of domain adaption as motivation into the abstract and intro which is missing in this version. I recommend acceptance of this paper as marginally above.


----- after the authors' response

Due to the raised concerns by another reviewer, I have reduced my score.

---

> ### Author Response · Authors · 2022-11-15
> **Response to Reviewer fvrm (1/2)**
>
> Dear reviewer,
>
> We thank you for your thorough review, and for your suggestions for improving our paper. We kindly invite you to look our revised manuscript and our general statement for a summary of changes.
>
> > The paper lacks clear motivation in the abstract and introduction. The organization of the paper makes it hard to distinguish between what is prior work and their work. For example, section 2 says preliminaries, but the authors mention "we propose ..." several times within this section. Please clarify.
>
> We decided to rewrite our introduction, and change the tone in our preliminaries for better clarity, and so as to better highlight our contributions.
>
> > How this work differs from the two prior works mentioned in the contribution paragraph?
>
> Bonneel et al., (2016) proposed a methodology for regressing the weights $\boldsymbol{\alpha}$ in the barycentric operator $\mathcal{B}(\boldsymbol{\alpha};\mathcal{P})$; Schmitz et al., (2018) proposed using $\mathcal{B}(\boldsymbol{\alpha};\mathcal{P})$ for performing Wasserstein Dictionary Learning. __These approaches are limited to data in the form of histograms__. This limits the applicability of these techniques in some areas of machine learning, for instance domain adaptation, where features live in high dimensions, making it untractable numerically. As such, our work proposes a dictionary learning framework on empirical distributions understood as point clouds. This enables applications to domain adaptation.
>
> In terms of implementation, our dictionary computation relies on a Free-Support Wasserstein Barycenter rather than a Fixed-Support Wasserstein Barycenter (see Cuturi and Doucet (2014) for further discussion). This raises different algorithmic challenges. For instance, to make the training tractable for large high dimensional datasets, we propose to sample both the atoms and the training data in each batch, which is converges empirically, and calls for new theoretical insights.
>
> Another aspect of novelty in our works is learning a dictionary of feature-label joint distributions $\hat{P}_{k}(X,Y)$. This involves integrating several aspects of previous works, such as class-based regularization of Courty et al., (2017) (which in our work we integrated in the ground-cost) and the label propagation proposed by Redko et al. (2019).
>
> In our revised manuscript we stated more clearly the points of novelty and difference between our framework, and that of Bonneel et al., (2016) and Schmitz et al., (2018).
>
> > The paper lacks a clear motivation. Abstract has no information on what problem this framework may solve and their motivation for pursuing such a model. Why someone should use their method? For example, based on the preliminaries subsection 2.2, is their model trying to address iid assumption made by the majority of the models that may not be valid on a certain application or during the test? This comment of mine is mainly to improve the clarity of the paper.
>
> In our experiemnts we explore the capabilities of our dictionary learning framework (e.g. dictionaries over gaussian distributions and color histograms), as well as (multi-source) domain adaptation. This latter case is indeed when one does not have an i.i.d. scenario over the training set. In our revised manuscript, we make more clear that we are proposing a new framework for (multi-source) domain adaptation, even though we believe our approach can be applied in other areas.
>
> > Are WJDOT and WBT SOTA? or they are OT-based SOTA? Please clarify.
>
> WJDOT and WBT are SOTA for Multi-Source Domain Adaptation. These methods employ OT as well. We stated this more clearly in our experiments sections in the revised manuscript (e.g. Sections 4.1.1. and 4.1.2.)
>
> > What is the complexity of the proposed method compared to other OT methods? In addition, how about a speed comparison?
>
> We provide a brief speed comparison in our experiments concerning color histograms (origianlly section 4.3., now appendix B.2.1). As for the complexity of our algorithm, let $N$ be the number of distributions in $\mathcal{Q}$, $K$ be the number of atoms, $N_{itb}$ be the number of maximum iterations of the Wasserstein Barycenter (Algorithms 2 or 3 in appendix A.2), and $n$ be the number of samples on the support of each $\hat{P}\_{k}$, and for $M$ batches of size $n\_{b}$ be the size of a batch. W.r.t. algorithm 1,
>
> __Complexity of a single training loop:__ $\mathcal{O}(N(N\_{itb} + M) n\_{b}^{3}\log n\_{b}),$
>
> __Complexity of computing the WED__
>
> 1. Precomputing the pairwise atom distances: $\mathcal{O}(K^{2}n^{3}\log n)$
> 2. Solving Eq. 10 (original paper)/ Eq. 11 (revised manuscript): $\mathcal{O}(K^{3}\log K)$
> 3. Overall: $\mathcal{O}(K^{2}(K\log K + n^{3}\log n))$
>
> (1) needs to be calculated once, at the end of DaDiL training loop, and $K \ll n$ in general (e.g., $K$ is on the order of tens whereas $n$ is on the order of thousands). Therefore solving (1) dominates over (2).
>
> __Continues in the next comment__

---

> > ### Author Response · Authors · 2022-11-15
> > **Response to Reviewer fvrm (2/2)**
> >
> > For new distributions (e.g., $\hat{Q} \not\in \mathcal{Q}$, $\mathcal{Q}$ used for fitting the dictionary), one can still find its embedding $\boldsymbol{\alpha}$ through Algorithm 1. with fixed atoms.
> >
> > > Please include C in Table 1 (right). Why is it not included?
> >
> > W.r.t. the original manuscript, Table 1 (left) and Table 1 (right) show experiments on different benchmarks. Table 1 (right) shows our experiments on the Office-31 dataset, which does not include the Caltech ( C ) domain. This is why C is not included.
> >
> > In our revised manuscript, we separated these two tables into Tables 1 and 2, so as not to confuse readers. We further clearly state that the Caltech-Office 10 dataset is the intersection of the Caltech 256 dataset of Griffin et al. (2007), and the Office 31 dataset of Saenko et al. (2010), that is, the (C) domain comes from the Caltech 256 dataset.
> >
> > > The ensemble method is not very clear on how it is used for reconstruction. Please elaborate.
> >
> > Upon learning a dictionary on $\mathcal{Q} = \set{\hat{Q}\_{S\_{\ell}}(X,Y)}\_{\ell=1}^{N\_{S}} \cup \set{\hat{Q}\_{T}(X)}$, our dictionary is composed by $K$ distributions $\set{\hat{P}\_{k}(X,Y)}\_{k=1}^{K}$, with samples $\set{(\mathbf{x}\_{i}^{(P\_{k})}, y\_{i}^{(P\_{k})})}\_{i=1}^{n}$, and weights $\mathcal{A}=\set{\boldsymbol{\alpha}\_{\ell}}_{\ell=1}^{N\_{S}+1}$. Since $\hat{Q}\_{N\_{S}+1} = \hat{Q}\_{T}$, by definition, one has $\boldsymbol{\alpha}\_{N\_{S}+1} = \boldsymbol{\alpha}\_{T}$. We thus learn $K$ classifiers, one on each atom through
> >
> > $$\hat{h}\_{k} = \underset{h\in\mathcal{H}}{\text{argmin}}\text{ }\hat{\mathcal{R}}\_{P\_{k}}(h)=\dfrac{1}{n}\sum\_{i=1}^{n}\mathcal{L}(h(\mathbf{x}\_{i}^{(P\_{k})}),y\_{i}^{(P\_{k})}),$$
> >
> > For predicting on the target distributions, we feed $\mathbf{x}\_{j}^{(Q\_{T})}$ to each $\hat{h}\_{k}$, then re-weight its predictions using the weights $\boldsymbol{\alpha}\_{N\_{S}+1} = \boldsymbol{\alpha}\_{T}$,
> >
> > $$\hat{h}\_{\boldsymbol{\alpha}\_{T}}(\mathbf{x}\_{j}^{(Q\_{T})}) = \sum_{k=1}^{K}\alpha\_{T,k}\hat{h}\_{k}(\mathbf{x}\_{j}^{(Q\_{T})}).$$
> >
> > We rewrote our discussion in order to clearly state these two strategies (e.g., Section 3.1. in our revised manuscript).
> >
> > > Is there a constraint on the dictionary in optimization formulation (1)?
> >
> > For histogram data (e.g., Schmitz et al., (2018)), i.e., $\mathbf{x}\_{\ell} \in \Delta\_{d}$, the atoms $\mathcal{P} \in (\Delta\_{d})^{K}$ and weights $\mathcal{A}\in(\Delta\_{K})^{N}$ are histograms as well. More generally, one can include regularization terms $\Omega\_{P}$ for atoms, and $\Omega\_{A}$ for representations. We now state these explicitly in our revised manuscript.
> >
> > > What are the dictionaries in Figure 5 left (original manuscript)? Possible to visualize them on the same figure?
> >
> > Thank you for suggesting this point. Our revised manuscript now includes the learned atoms in Figure 4 ( c ).
> >
> > > Is the method in Figure 3b (original manuscript) DaDiL-E and R?
> >
> > In Figure 3b (original manuscript) we show the learned atoms (scatterplot) and the learned classifiers (heatmap). We included this description in the caption of the corresponding figure in our revised manuscript (Figure 1)

---

> > > ### Comment · Reviewer_fvrm · 2022-12-01
> > > **Revised Review Score**
> > >
> > > I thank the authors for their responses and modifications to the paper. Given the raised concerns by Reviewer 6SsD on the correctness of one of the theorems, I have reduced my score and do not recommend an acceptance. I strongly suggest the authors address the concerns and re-submit to a future conference.

---

### Official Review · Reviewer_6SsD · 2022-11-02

**Confidence:** 3
**Correctness:** 1
**Technical Novelty And Significance:** 1
**Empirical Novelty And Significance:** 2
**Recommendation:** 3

**Clarity, Quality, Novelty And Reproducibility:**

See above.

Writing feedback:
- Footnote on page 4 seems like a quote, if so it should be in quotes
- Sum in equation 9 should be over $\ell$ not $i$
- Algorithm 1 is written in a confusing manner, particular the "sample" lines. As I understand it, you are sampling only at the beginning (before the for loop begins) and then moving the support points along gradient descent. This should be made more clear
- Should have parentheses around $1/L$ in line 7 of Algorithm 1
- Notation is generally confusing and not clearly defined (if defined at all): for example in statement of Theorem 3.2, $\ell$ is not quantified, and the $\alpha$'s are not defined. Even the $\alpha^*$ is not clearly defined, as it depends on $\hat{Q}$ but that is not made explicit. Or also equation 13 involves undefined terms and no explicit high-probability quantification (should read "... holds with probability at least $1 - \delta$)
- Figure 5 has no explanation of what the loss is
- $\pi^c$ in first inequality on page 16 should be $\pi^2$
- Section 3.2 could use more explanation of what the actual method of producing the labels is, if only a reference back to equation 8 in section 2. I found that part confusing

**Strength And Weaknesses:**

Strengths:
- Studies an important problem, that of domain adaptation with optimal transport
- Proposes a new method based on using Wasserstein barycenters to form dictionaries of a larger set of point distributions
- Achieves some state of the art empirical results for multi-class domain adaptation
- Many parts are well-written, despite the fact that much mathematical notation is not defined before it is used

Neutral:
-While there is some mathematical justification given for their method, it doesn't require novel theory to accomplish. Also, there are some issues with correctness

Weaknesses:
- The barycentric coordinates definition is equivalent to 1NN: the minimizer in equation 11 is simply just a point mass on the $\hat{P}_k$ which is closest to $\hat{Q}$. The definition in Bonneel et al is distinct from yours, because it also includes a minimization over $\hat{Q}$
- The proof of Theorem 3.2 is incorrect: the last inequality on page 16 goes in the opposite direction, so the proof of Theorem 3.2 as written is incorrect
- Figure 3 doesn't make sense because it is not learning a dictionary since $K = N$ there
- It isn't clearly explained how the $alpha_T$ weights for the target distribution $\hat{Q}_T$ are computed

Bonneel, N., Peyré, G., & Cuturi, M. (2016). Wasserstein barycentric coordinates: histogram regression using optimal transport. ACM Trans. Graph., 35(4), 71-1.

**Summary Of The Paper:**

This work develops a novel method for applying Wasserstein barycenters to learn a dictionary for a collection of point clouds. Their method works by optimizing an objective over the support of the dictionary distributions and the dictionary weights with a Wasserstein barycenter loss. In particular, the cost function in the labeled data case forces points with the same labels to be mapped to one another. They apply this method to the problem of multi-class domain adaptation where they use the learned dictionary to transfer labels from a target dataset to an un-labeled dataset, and achieve some state-of-the-art results.

**Summary Of The Review:**

This work studies an important problem and proposes a fairly natural and yet still novel solution. The empirical performance of their solution is strong. However, their theoretical results have some correctness issues (particularly Theorem 2), and otherwise some of their presentation (particularly the statements of their theoretical results, as well as the definitions of their classification procedures in section 3.2) are opaque. For these reasons, I think their work is just below the criteria of acceptance.

---

> ### Author Response · Authors · 2022-11-15
> **Response to Reviewer 6SsD (1/1)**
>
> Dear reviewer,
>
> We thank you for your thorough review, and for taking the time to read our appendices. We also thank your remark w.r.t Theorem 3.2.; our proof was indeed incorrect and we address it below. We kindly invite you to look our revised manuscript and our general statement for a summary of changes.
>
> > The barycentric coordinates definition is equivalent to 1NN: the minimizer in equation 11 (original manuscript) is simply just a point mass on the $\hat{P}_{k}$ which is closest to $\hat{Q}$. The definition in Bonneel et al is distinct from yours, because it also includes a minimization over $\hat{Q}$.
>
> Thank you for this point. In the first version of our manuscript we indeed wrote the barycentric coordinates definition wrongly. It should read, in our notation, as
>
> $$ \boldsymbol{\alpha}^{\star} = \text{argmin}\_{\boldsymbol{\alpha}\in\Delta\_{K}}W\_{c}(\mathcal{B}(\boldsymbol{\alpha};\mathcal{P}),\hat{Q}), $$
>
> This mistake was corrected as you can see in Eq. 12 of our revised manuscript.
>
> > The proof of Theorem 3.2 is incorrect: the last inequality on page 16 goes in the opposite direction, so the proof of Theorem 3.2 as written is incorrect.
>
> Thank you for pointing out our mistake. Indeed, our proof was incorrect. The correct reasoning should be as follows,
>
> $$
> \sum\_{k\_{1}=1}^{K}\sum\_{k_{2}=1}^{K}\pi\_{k\_{1},k\_{2}}W\_{c}(\hat{P}\_{k\_{1}},\hat{Q}\_{1}) = \sum\_{k\_{1}=1}^{K}\alpha_{1,k\_{1}}W\_{c}(\hat{P}\_{k\_{1}},\hat{Q}\_{1}) \leq \sum\_{k=1}^{K}\alpha_{1,k}(W\_{c}(\hat{P}\_{k},\hat{B}\_{1}) + W\_{c}(\hat{B}\_{1},\hat{Q}\_{1}))
> $$
>
> where we used the triangle inequality. As follows,
>
> $$
> \sum_{k_{1}=1}^{K}\sum\_{k\_{2}=1}^{K}\pi\_{k\_{1},k\_{2}}W\_{c}(\hat{P}\_{k\_{1}},\hat{Q}\_{1}) \leq  W\_{c}(\hat{B}\_{1},\hat{Q}\_{1}) + \sum_{k=1}^{K}\alpha\_{1,k}W\_{c}(\hat{P}\_{k},\hat{B}\_{1}).
> $$
>
> Following the original reasoning of A.3., we have a different bound,
>
> $$
>     \text{WED}(\hat{Q}\_{1},\hat{Q}\_{2}) \leq W\_{c}(\hat{Q}\_{1},\hat{Q}\_{2}) + \underbrace{\sum\_{k=1}^{K}\alpha\_{1,k}W\_{c}(\hat{P}\_{k},\hat{B}\_{1}) + \sum\_{k=1}^{K}\alpha\_{2,k}W\_{c}(\hat{P}\_{k},\hat{B}\_{2})}\_{\text{Dictionary Geometry}} + \underbrace{W\_{c}(\hat{B}\_{1},\hat{Q}\_{1})+W\_{c}(\hat{B}\_{2},\hat{Q}\_{2})}_{\text{Reconstruction Error}}.
> $$
>
> > Figure 3 (original manuscript) doesn't make sense because it is not learning a dictionary since $K=N$ there
>
> In Figure 3 (original manuscript) we learn a labeled dictionary $\set{\hat{P}\_{k}(X,Y)}\_{k=1}^{4}$. This problem is not trivial, because we have one __unlabeled__ distribution in $\mathcal{Q}$. For better clarity, and as not to mislead readers, we reworked this example using fewer atoms, as is shown in Figure 1 of our revised manuscript.
>
> > It isn't clearly explained how the $\boldsymbol{\alpha}_{T}$ weights for the target distribution $\hat{Q}_{T}$ are computed
>
> We consider $\mathcal{Q} = \set{\hat{Q}\_{S\_{\ell}}(X,Y)}\_{\ell=1}^{N\_{S}} \cup \set{\hat{Q}\_{T}(X)}$ as the "learning data" in our dictionary learning problem, which means that $\mathcal{A} = \set{\boldsymbol{\alpha}\_{\ell}}\_{\ell=1}^{N_{S}+1}$, that is $\boldsymbol{\alpha}\_{T} = \boldsymbol{\alpha}\_{N\_{S}+1}$. We now explicitly state this in our revised manuscript (section 3.1. Domain Adaptation).
>
> > Footnote on page 4 seems like a quote, if so it should be in quotes
>
> For the sake of clarity, we removed the quotation.
>
> > Algorithm 1 is written in a confusing manner, particular the "sample" lines. As I understand it, you are sampling only at the beginning (before the for loop begins) and then moving the support points along gradient descent. This should be made more clear
>
> We actually sample __inside__ the for-loop, as at each iteration we select a mini-batch from each atom $\hat{P}\_{k}$, and a mini-batch from each distribution $\hat{Q}\_{\ell}$. We will rewrite the algorithm so that it better resembles what is implemented in practice (in Pytorch).
>
> > Notation is generally confusing and not clearly defined (if defined at all): for example in statement of Theorem 3.2, $\ell$ is not quantified, and the $\alpha$'s are not defined. Even the $\alpha^{*}$ is not clearly defined, as it depends on $\hat{Q}$ but that is not made explicit. Or also equation 13 involves undefined terms and no explicit high-probability quantification (should read "... holds with probability at least $1-\delta$)
>
> Thank you for pointing our these issues. We corrected them in our revised manuscript.

---

> > ### Comment · Reviewer_6SsD · 2022-11-23
> > **More bugs in your proofs**
> >
> > Thanks for your revisions, but I have now found even more bugs in your proofs, particularly that of Theorem 3.2
> >
> > First of all, the $\alpha$ in barycentric coordinates is not shown to exist. It is claimed that this $\alpha$ is the solution of a convex optimization problem but I am virtually certain that the problem is not convex (and indeed in Bonneel et al they say it is not). So rigorous results about the Wasserstein Embedding Distance are impossible without resolving this issue.
> >
> > Leaving aside the issues of existence and uniqueness, the statement of Theorem 3.2 is obvious because it is equivalent to a Wasserstein distance on the barycentric coordinates $\alpha$, which is a metric, and so it is indeed a pseudo-metric when considered on the original distributions. But the proof of Theorem 3.2 is still wrong! In particular the proof of the triangle inequality is completely wrong, as it uses optimality for $\pi^{(1)}$ to $\pi^{(2)}$, but $\pi^{(1)}$ is only optimal when compared to other couplings with the same marginals.
> >
> > I am very concerned about the correctness of the remaining theoretical "results", and about the care with which the paper was written overall. I do not think it is ready for publication.

---

### Official Review · Reviewer_LdfT · 2022-11-04

**Confidence:** 4
**Correctness:** 3
**Technical Novelty And Significance:** 2
**Empirical Novelty And Significance:** 1
**Recommendation:** 3

**Clarity, Quality, Novelty And Reproducibility:**

The paper is not clear.

The quality IMHO is below the expectation of an ICLR paper.

I didn't find novelty, but that might be because the paper is hard to read.

The results in Sec 4 seem reproducible given the description of the algorithm.

**Strength And Weaknesses:**

Strength:

The bounds in Theorem 3.1, 3.2, and 3.3 are novel to me.

Exploring Wasserstein dictionary learning for multi-source domain adaptation is a direction worth to explore.


Weaknesses

The paper does not offer much beyond what Schmitz et al. did in their Wasserstein Dictionary Learning paper. At least it's not clear to me the contribution of this paper.


The paper is hard to read and understand.

* 1. Introduction is basically a historic review of OT which is not informative and mostly irrelevant to the scope of the paper.
* 2. Preliminaries are mathematical basics plus some history for dictionary learning, domain adaptation, and OT. They are a collection of previous knowledge in these areas and authors presented them without a flow in between. Some history and citations are stalling readers and are redundant for readers to understand the rest of the paper.

* 3.1 Wasserstein Embedding distance as a proxy for the empirical Wasserstein distance. I appreciate authors' effort on Theorem 1 & 2 but I don't follow the intension of this proxy. I didn't find it being used in the rest of the paper and it doesn't seem to be a theoretical foundation for 3.2 either.

* 3.2 is not clear. Authors jumped too quickly from problem statement to justifying their solution. No reasoning is given linking the problem and the solution. And it's not clear at least to me that how Theorem 3.3 which is yet another bound can justify "this strategy". "We want to learn a dictionary of labeled distributions": how do we use the dictionary for tasks on the target domain?

* 4.2 Because it's not clear in 3.2 how a dictionary is used for domain adaptation, it's hard to understand 4.2 except a bare table showing high numbers.

* 4.3 "In this experiment, we want to learn a dictionary over 50 artworks": I don't quite follow the reason to learn dictionaries of artworks as RGB images. "which shows that we are able to capture the Wasserstein geometry faithfully". This seems like a toy example for validation of the learned embeddings. I don't find it informative given we already have 4.1.

**Summary Of The Paper:**

The paper proposes using Wasserstein barycenters as a dictionary to recover multiple datasets and uses it for domain adaptation.
It shows that their approximation with the learned dictionary is bounded. Empirical results show that the proposed method marginally improved the DA classification accuracy.

**Summary Of The Review:**

I recommend rejection at this point. The paper does not offer new insights into understanding the connection between dictionary learning and OT and it's hard to read.

---

> ### Author Response · Authors · 2022-11-15
> **Response to Reviewer LdfT (1/2)**
>
> Dear reviewer,
>
> We thank you for your thorough review. We hope to clarify and address the issues raised in your review. In addition, we kindly invite you to look our revised manuscript and our general statement for a summary of changes.
>
> > The paper does not offer much beyond what Schmitz et al. did in their Wasserstein Dictionary Learning paper. At least it's not clear to me the contribution of this paper.
>
> The work of Schmitz et al. (2018) is __limited to distributions in the form of histograms__. This makes their approach numerically untractable when the dimensionality of data is large (e.g. Domain Adaptation, where features live in spaces of dimensionality ~$10^{3}$). In this sense we offer a novel framework, considering dictionaries over empirical distributions. We consider this approach novel, as dictionary learning is commonly applied over vectors/histograms, rather than distributions of samples. This formulation comes in hand for (multi-source) domain adaptation, as we showed in our theoretical discussion and experiments.
>
> However, we do agree that this was not clearly discussed in the first version of our paper. While revising our manuscript, we made the effort to make our contribution more evident to readers.
>
> > Introduction is basically a historic review of OT which is not informative and mostly irrelevant to the scope of the paper. Preliminaries are mathematical basics plus some history for dictionary learning, domain adaptation, and OT. They are a collection of previous knowledge in these areas and authors presented them without a flow in between. Some history and citations are stalling readers and are redundant for readers to understand the rest of the paper.
>
> We thank the reviewer for pointing out issues with our discussion. In the revised version of our paper, we rewrote the introduction to highlight our approach's novelty and that our primary field of application is domain adaptation.
>
> > Wasserstein Embedding distance as a proxy for the empirical Wasserstein distance. I appreciate authors' effort on Theorem 1 & 2 but I don't follow the intension of this proxy. I didn't find it being used in the rest of the paper and it doesn't seem to be a theoretical foundation for 3.2 either.
>
> We use the WED in our toy example concerning Gaussians (originaly section 4.1., now section 4.2.) and when learning dictionaries over color histograms (originaly section 4.2., now appendix B.2.1). At this point in our experimentation, we find that the WED has mainly two purposes.
>
> - (i) it defines a novel semi-metric that has interesting properties on its own (e.g., Thm. 1 & 2)
> - (ii) it allows a faster comparison of large-scale empirical distributions.
>
> Point (i) is explored in our toy example about gaussians. Point (ii) is explored in the color histogram example, where we were able to calculate the distance between artworks without sub-sampling the support of its color histograms, and in overall DaDiL (and thus calculating the WED) is faster than calculating pairwise Wasserstein distances.
>
> > 3.2 is not clear. Authors jumped too quickly from problem statement to justifying their solution. No reasoning is given linking the problem and the solution. And it's not clear at least to me that how Theorem 3.3 which is yet another bound can justify "this strategy". "We want to learn a dictionary of labeled distributions": how do we use the dictionary for tasks on the target domain?
>
> When using DaDiL for domain adaptation, we consider learning a dictionary over the set $\mathcal{Q} = \set{\hat{Q}\_{S\_{\ell}}(X,Y)}\_{\ell=1}^{N\_{S}}\cup\set{\hat{Q}\_{T}(X)}$, meaning that the atoms $\mathcal{P}=\set{\hat{P}\_{k}(X,Y)}\_{k=1}^{K}$, and weights $\mathcal{A}=\set{\boldsymbol{\alpha}\_{\ell}}\_{k=1}^{N_{S}+1}$ are learned using __labeled__ samples from each source distribution $\hat{Q}\_{S\_{\ell}}$ (i.e., $\set{(\mathbf{x}\_{i}^{(Q\_{S\_{\ell}})},y\_{i}^{(Q\_{S\_{\ell}})})}\_{i=1}^{n\_{Q\_{S\_{\ell}}}}$), and __unlabeled__ samples from the target $\hat{Q}\_{T}$ (i.e., $\set{\mathbf{x}_{i}^{(\hat{Q}\_{T})}}\_{i=1}^{n\_{Q\_{T}}}$). In our paper we use the learned dictionary in 2 ways DaDiL-R (for reconstruction) and DaDiL-E (for ensembling).
>
> For __DaDiL-R__ we use the reconstruction $\hat{B}\_{T}$ of $\hat{Q}\_{T}$, obtained through the Wasserstein barycenter of $\mathcal{P}=\set{\hat{P}_{k}(X,Y)}\_{k=1}^{K}$, $\hat{B}\_{T} = \mathcal{B}(\boldsymbol{\alpha}\_{T};\mathcal{P})$. This distribution has as support,
>
> $$\mathbf{X}^{(B\_{T})} = n\sum_{k=1}^{K}\alpha\_{k}\pi\_{k}^{T}\mathbf{X}^{(P\_{k})}\text{ and }\mathbf{Y}^{(B\_{T})}=n\sum_{k=1}^{K}\alpha\_{k}\pi\_{k}^{T}\mathbf{Y}^{(P\_{k})},$$
>
> determined through the fixed-point iterations of ́Alvarez-Esteban et al. (2016).
>
> __Continues in the next comment__

---

> > ### Author Response · Authors · 2022-11-15
> > **Response to Reviewer LdfT (2/2)**
> >
> > For __DaDiL-E__, we learn a classifier for each atom, i.e.,
> >
> > $$\hat{h}\_{k} = \underset{h \in \mathcal{H}}{\text{argmin}}\text{ }\hat{\mathcal{R}}\_{P\_{k}}(h)=\dfrac{1}{n}\sum\_{i=1}^{n}\mathcal{L}(h(\mathbf{x}\_{i}^{(P\_{k})}),y\_{i}^{(P\_{k})}),$$
> >
> > for samples $\mathbf{x}\_{j}^{(Q\_{T})} \sim Q\_{T}$ we apply each $\hat{h}\_{k}$ and weight its predictions using $\boldsymbol{\alpha}\_{T}$,
> >
> > $$h\_{\boldsymbol{\alpha}\_{T}}(\mathbf{x}\_{j}^{(Q\_{T})}) = \sum\_{k=1}^{K}\alpha\_{T,k}\hat{h}\_{k}(\mathbf{x}\_{j}^{(Q\_{T})})$$
> >
> > We rewrote our discussion in order to clearly state these two strategies (e.g., Section 3.1. in our revised manuscript).
> >
> > > 4.3 "In this experiment, we want to learn a dictionary over 50 artworks": I don't quite follow the reason to learn dictionaries of artworks as RGB images. "which shows that we are able to capture the Wasserstein geometry faithfully". This seems like a toy example for validation of the learned embeddings. I don't find it informative given we already have 4.1.
> >
> > In this experiment, we wanted to illustrate the properties of our dictionary, and exemplify the ideas of atoms and the embeddings they generate on more empirical distributions more realistic than Gaussians. We also wanted to illustrate how the WED can be beneficial, in terms of calculation w.r.t. the Wasserstein distance. In this experiment, each artwork has millions of pixels (e.g., $n = h \times w$, for an image of height $h$ and width $w$). For the exact Wasserstein distance computation to be tractable, we need to sub-sample the target distributions support to $n = 2048$. Our dictionary thus allows us to define a pseudo-metric without the need to sub-sample the distributions support.

---

### Author Response · Authors · 2022-11-15
**Overview of changes to our submission**

Dear Reviewers,

We thank you for your thorough reviews. We took the time to make a revised manuscript, taking into account your comments, suggestions and remarks for clarifications. We make this general statement so as to summarize what was changed from the first, to the second version of our paper.

## Introduction

- We rewrote the introduction. As pointed out by reviewer LdfT, some of the historical account and references were stalling readers. We took that into account for making a more straightforward discussion.
- We state more enfatically the novelty of our work. This was pointed out by all reviewers, and we do agree that in the first version of our paper, we did not make explicit in what our framework differs from previous work.
- We explicit that our paper seeks to solve/contribute to __domain adaptation__. This was somewhat unclear in the first version.
- We added a brief description of the notation used.

## Preliminaries

- We re-organized our background section, so as to create have a more fluid discussion. It now reads as OT -> DA - > Dictionary Learning.
- We removed sentences containing "We propose ..." in the preliminaries.

## Dataset Dictionary Learning

- We expanded this section, as reviewers found that we jumped too quickly between our proposed method and the experiments.
- We changed the order of exposition. It now reads as Dataset Dictionary Learning/Domain Adaptation/A proxy for the Empirical Wasserstein Distance
    - Due to the order change, the order of theorems was changed as well. Our bounds for DA using DaDiL-E corresponds to Theorem 3.1., whereas theorems about the WED correspond to 3.2. and 3.3.
- When describing Dataset Dictionary Learning, we highlight the differences with previous work.
- In Section 3.1. Domain Adaptation, we define more explicitly DaDiL-R and DaDiL-E. For DaDiL-R, we give pointers to the equations that allow us find $\mathbf{X}^{(B\_{T})}$ and $\mathbf{Y}^{(B\_{T})}$. More specifically, we highlight that we find target domain weights $\boldsymbol{\alpha}\_{T}$ by considering $\hat{Q}\_{N\_{S}+1}=\hat{Q}\_{T}$. For DaDiL-E we explicitly mention that it is trained on each atom, and that target domain predictions are obtained by weighting atom classifiers predictions on target domain features.
- In Section 3.2. A Proxy for the Empirical Wasserstein Distance, a few issues were corrected
    - Reviewer 6SsD pointed out that the definition of barycentric coordinates regression was wrong. We corrected it as shown in Eq. 12.
    - Reviewer 6SsD pointed out that our proof of Theorem 3.3. (previously 3.1.) was incorrect. We corrected it and commented on the bound we managed to find.

## Experiments

- We moved our example of Color Histogram learning into the appendix. While the example is interesting, we found that Domain Adaptation merits a bigger place in our experimentaiton.
- We changed the order of DA (now 4.1.) and Learning the Manifold of Gaussian distributions (now 4.2.).
- We broke the DA section into 3 subsections: 4.1.1. Shallow DA (experiments on Caltech-Office 10) and 4.1.2. Deep DA (experiments on Office 31). __A new sub-section was added__ (4.1.3.), showing adaptation performance with atom interpolations, i.e. $\hat{B} \in \set{\mathcal{B}(\boldsymbol{\alpha};\mathcal{P}):\boldsymbol{\alpha}\in \Delta\_{K}}$
- In our example with Gaussians, we added a few Figures,
    - The atoms learned with our dictionary are now shown in Figure 4 (c). We included the correlation analysis of the WED and $W_{2}$ between distributions in the manifold of Gaussian distributions (Figure 4 (f)).

---

### Decision · Program_Chairs · 2023-01-20

**Decision:**

Reject

**Justification For Why Not Higher Score:**

The paper has clarity issues, both in terms of notation and motivation/contributions. The submitted version also contained technical errors (incorrect mathematical claims).

**Justification For Why Not Lower Score:**

N/A

**Metareview: Summary, Strengths And Weaknesses:**

The paper develops algorithms for learning dictionaries over datasets (here, point clouds), and applies them to unsupervised domain adaptation problems. The dictionary learning problem is formulated as one of minimizing a Wassserstein loss, subject to sparsity regularization. The learned dictionary is applied to domain adaptation through two strategies — an ensemble strategy in which atoms have associated classifiers, and a label propagation strategy, in which labels are transported via the learned transport plan. The paper presents experiments on domain adaptation on two datasets.

Reviewers found the problem of domain adaptation with large point clouds to be well-motivated — in particular, proposed methods are more scalable than previous Wasserstein dictionary learning methods based on histograms. At the same time, several reviewers found the paper unclear: both in terms of mathematical notation and definitions, as well as in terms of high-level motivations and contributions. Moreover, the review raised several issues with the correctness of the submitted version (at least one mathematical claim was incorrect). While the paper has potential merit as a scalable approach to Wasserstein dictionary learning, in current form the consensus is to reject.